# High-throughput immune repertoire analysis with IGoR

Quentin Marcou[1], Thierry Mora[2] & Aleksandra M. Walczak[1]

High-throughput immune repertoire sequencing is promising to lead to new statistical diagnostic tools for medicine and biology. Successful implementations of these methods require a correct characterization, analysis, and interpretation of these data sets. We present IGoR (Inference and Generation Of Repertoires)—a comprehensive tool that takes B or T cell receptor sequence reads and quantitatively characterizes the statistics of receptor generation from both cDNA and gDNA. It probabilistically annotates sequences and its modular structure can be used to investigate models of increasing biological complexity for different organisms. For B cells, IGoR returns the hypermutation statistics, which we use to reveal co-localization of hypermutations along the sequence. We demonstrate that IGoR outperforms existing tools in accuracy and estimate the sample sizes needed for reliable repertoire characterization.

[1] Laboratoire de Physique Théorique, CNRS, Sorbonne Université and École Normale Supérieure (PSL), 24, Rue Lhomond, 75005 Paris, France. [2] Laboratoire de Physique Statistique, CNRS, Sorbonne Université, Université Paris-Diderot, and École normale supérieure (PSL), 24, Rue Lhomond, 75005 Paris, France. Thierry Mora and Aleksandra M. Walczak contributed equally to this work. Correspondence and requests for materials should be addressed to T.M. (email: tmora@lps.ens.fr) or to A.M.W. (email: awalczak@lpt.ens.fr)

The adaptive immune system recognizes pathogens by binding their antigens to specific surface receptors expressed on T and B cells. The recent advent of high-throughput immune repertoire sequencing (RepSeq)[1–4] gives us direct insight into the diversity of B cell and T cell receptor (BCR and TCR) repertoires with great potential to change the way we diagnose, treat, and prevent immune system-related disorders. A growing number of algorithms and software tools have been designed to address the new challenges of RepSeq, in particular sequence analysis, germline assignment and clone construction[5–10]. However, each receptor sequence can be generated in a large number of ways, which we call "scenarios," comprising the processes leading to pre-selection receptors: recombination of germline segments, insertions and deletions, and hypermutations. Different germline segments can recombine with each other with different frequencies, and the number of insertions and deletions is random, so that the overall receptor generation process cannot be described deterministically. Standard assignments introduce systematic errors when describing this inherently stochastic process. Quantitatively characterizing the diversity and the biases of these mechanisms remains a challenge for understanding adaptive immunity and applying RepSeq for diagnostics.

We present a flexible computational method and software tool, IGoR (Inference and Generation of Repertoires), that processes raw immune sequence reads from any source (cDNA or gDNA) and learns unbiased statistics of V(D)J recombination and somatic hypermutations. Using these statistics, for each sequence IGoR outputs a whole list of potential recombination and hypermutation scenarios, with their corresponding likelihoods. IGoR's performance at identifying the correct VDJ recombination scenario is two times better than current state-of-the-art methods. IGoR used as a sequence generator produces an arbitrary number of randomly rearranged sequences with the same statistics as in the data set. Applied to BCRs, IGoR learns a context-dependent hypermutation model to identify hotspots, which allows for a comprehensive analysis of the mutational landscape of BCRs.

## Results

**Probabilistic assignment of recombination scenarios.** V(D)J recombination selects two or three germline segments (Variable —V and Joining—J loci for TCR α and BCR light chains; and the V, Diversity—D, and J loci for TCR β and BCR heavy chains) from a library of germline genes, and assembles them while deleting base pairs and inserting other non-templated ones at the junctions (Fig. 1a). B cell receptors can further diversify through somatic hypermutations during affinity maturation. The recombination process is degenerate, as the same sequence can be generated in many different ways[11]. IGoR takes as input a list of sequences obtained from the initial pre-processing of raw reads, controling for read quality, and grouping unique or very similar sequences together (as can be done using existing software such as MiXCR[8]). IGoR starts by listing the possible recombination and hypermutation scenarios leading to an observed sequence in the data set. It then assigns probability weights reflecting the likelihood of these scenarios. As the example in Fig. 1a shows, explored scenarios can be very different yet have comparable contributions to the sequence likelihood. Since exploring all possible scenarios would be computationally too costly, IGoR restricts its exploration to the reasonably likely ones (see Supplementary Note 5). Scenario exploration takes from 1 ms up to less than a second per sequence on a single CPU core, depending on the chain (see Table 1, and full distributions of runtimes in Supplementary Fig. 1). Different recombination architectures can be configured within IGoR by specifying dependencies between elementary events (gene choices, deletions, insertions, and hypermutations) through an acyclic-directed graph, or Bayesian network, as illustrated in Fig. 1b for the case of BCR heavy chains (see "Methods" section for the other used structures). Note that IGoR actually runs faster on longer reads, as this tends to lift ambiguities in the assignment of the V gene, decreasing summation time over scenarios.

IGoR functions according to three modes: learning, analysis, and generation (Fig. 1c). In the learning mode, IGoR infers the recombination statistics of large data sets of sequences using a

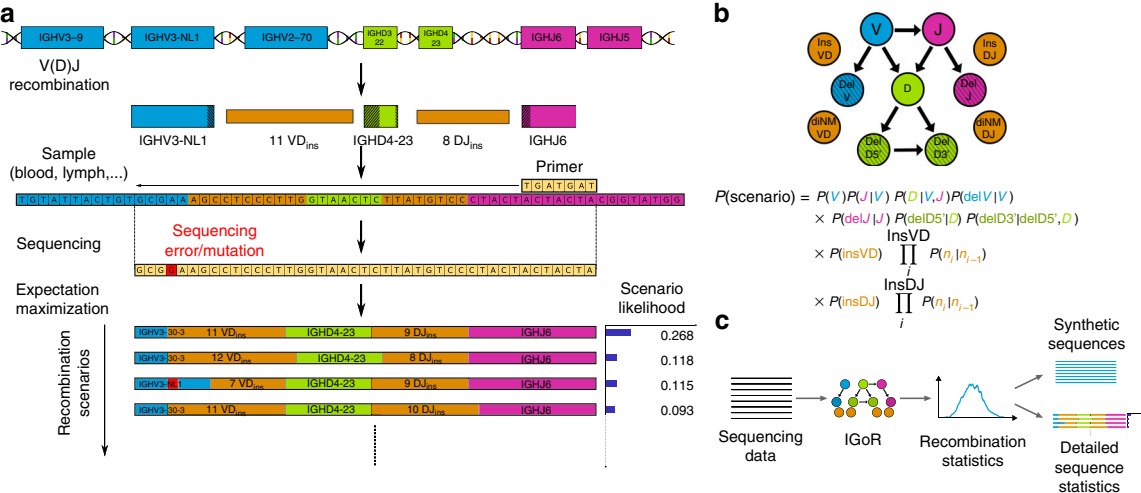

**Fig. 1** IGoR's pipeline for sequence analysis. **a** V(D)J recombination proceeds by joining randomly selected segments (V, D, and J segments in the case of TRB and IGH). Each segment gets trimmed at its ends (hashed areas), and a varying number of non-templated insertions are added between them (orange). Hypermutations (in the case of B cells) or sequencing errors (in red) further enhance diversity. IGoR lists putative recombination scenarios consistent with the observed sequence, and weighs them according to their likelihood. **b** The likelihood of each scenario is computed using a Bayesian network of dependencies between the recombination features (V, D, J segment choices, insertions, and deletions), as illustrated here for the human TRB locus. Architectures for TRA and IGH are described in Methods. **c** IGoR's pipeline includes three modes. In the learning mode, IGoR learns recombination statistics from data sequences. In the analysis mode, IGoR outputs detailed recombination scenario statistics for each sequence. In the generation mode, IGoR produces synthetic sequences with specified recombination statistics

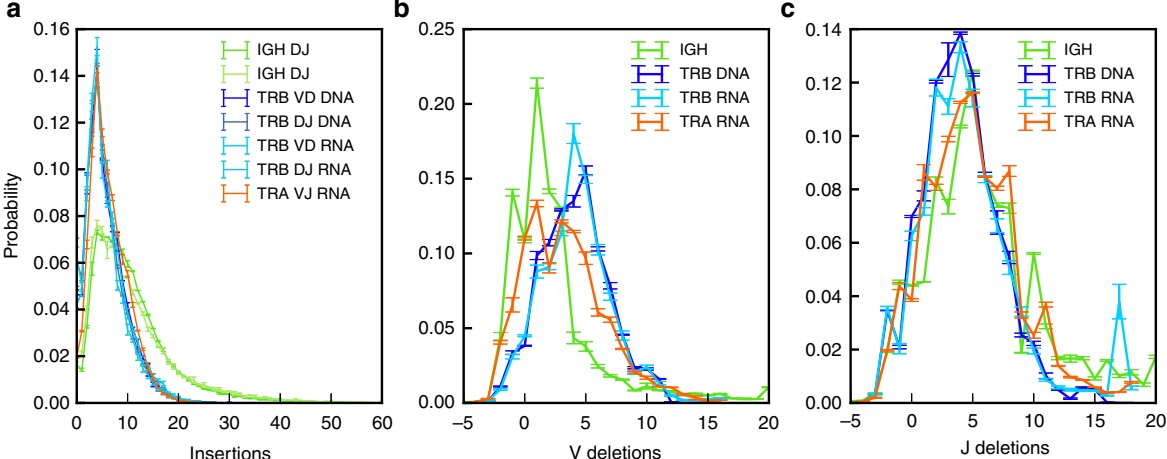

**Fig. 2** IGoR infers reproducible recombination statistics between individuals. **a** Distribution of the number of insertions at the junctions of recombined genes: IGH at the VD and DJ junctions from DNA data[14], TRB at the VD and DJ junction from both DNA[13] and mRNA data[12], and TRA at the VJ insertion site from mRNA data[12]. The insertion profile is assumed to be universal for all genes and the distributions are also reproducible between TRA and TRB. **b**, **c** Average distribution over all genes of the number of deletions across **b** V and **c** J genes. The gene-by-gene distributions of the most frequent genes are reported in Supplementary Fig. 5. Negative deletions correspond to palindromic insertions (P-nucleotides), e.g., −2 means 2 P-nucleotides. The inferred distributions are robust to the choice of individuals, genetic material (mRNA or DNA), and sequencing technology. Error bars show 1 standard deviation across individuals

sparse expectation-maximization algorithm (see "Methods" section). In the analysis mode, IGoR assigns recombination events to sequences in a probabilistic way, by outputting the most likely scenarios ranked by their probabilities, as well as the overall generation probability of the sequence. In the generation mode, IGoR outputs random sequences with specified statistics, e.g., learned from real data sets.

**Inference of V(D)J recombination**. We used IGoR's learning mode to infer the accurate statistics of V(D)J recombination from four data sets comprised of unique sequences of non-productive rearrangements of three different chains, sequenced either at the levels of mRNA (TCRα chain or "TRA," and TCRβ chain or "TRB"[12]) or DNA (TRB[13], and BCR immunoglobulin heavy chain or "IGH" from naive B cells[14]), generalizing earlier methods[15–17]. Restricting to non-productive unique sequences allowed us to avoid biases introduced by functional selection. For BCR, we only consider out-of-frame sequences from the naive repertoire, which have no hypermutation insertions and deletions, giving us confidence in their non-productive status. The expectation-maximization algorithm converged within a few iterations (see Supplementary Fig. 2 for the convergence of the likelihoods and of the insertion distributions of both TRB and IGH).

The same TRB insertion and deletion distributions were inferred regardless of the individual, laboratory of origin, or sequencing protocol, and of whether DNA (light blue distributions in Fig. 2) or mRNA (dark blue) was used. By contrast, V and J gene usage varied moderately but significantly across individuals, and even more across sequencing technologies, suggesting possible primer-dependent biases (Supplementary Fig. 3 for TRB, see also Supplementary Fig. 4 for IGH D–J gene usage). Insertions at the TRA V–J junction, and at the TRB V–D and D–J junctions have similar distributions (Fig. 2a), as previously reported[17]. IGH sequences have significantly more insertions at the junctions than TCRs, consistent with previous observations[16]. The statistics of deletions (Supplementary Fig. 5), and in particular negative deletions which by convention correspond to palindromic insertions, depend on gene segments for both BCR and TCR (Supplementary Fig. 6), updating previous estimates in IGH[18].

We then validated the learning algorithm on synthetic data sets. Sequences were generated in batches of $10^3$–$10^5$ by IGoR with a variable error rate, using statistics inferred from 60 bp DNA TRB data. The length of the synthetic sequences was chosen to match the experimentally analyzed read lengths. IGoR's learning algorithm was then run on these raw sequences, and the resulting statistics compared to the known ground truth. We found that the inference was highly accurate for data sets of $10^5$ sequences and an error rate set to its typical experimental value, $10^{-3}$ (Fig. 3a, b). However, not all high-throughput sequencing data sets reach this depth, especially when restricted to unique non-productive sequences. To assess how these limitations affect accuracy, we calculated the Kullback–Leibler divergence (a non-parametric measure of difference between probability distributions, see "Methods" section) between the true TRB distributions and the inferred ones, for varying sizes of data sets and error rates. For an error rate of $10^{-3}$, ~5000 unique out-of-frame sequences (which can be obtained from less than 2 ml of blood with current mRNA sequencing technologies[12]) were sufficient to learn an accurate model of TRB (Fig. 3c, see also Supplementary Fig. 7 for insertion and deletion distributions), with the majority of the estimation error due to deletion profiles (which account for the majority of parameters). The typical divergences, of the order of 1 bit, are small compared to the total entropy of the process, ~50 bits[19], suggesting absence of overfitting. Increasing the error rate has little effect up to rates of $10^{-2}$, but significantly degrades accuracy for larger error rates, $10^{-1}$ (Fig. 3d), with the gene usage distribution affected the most (Supplementary Fig. 8). In addition, hypermutation rates in BCRs, which IGoR treats in the same way as errors, can reach 1–10%, and they show a long tail with about 5% of sequences having a hypermutation rate of 30% or greater (Supplementary Fig. 9). This suggests that the recombination statistics of BCRs should preferably be inferred using sequences from naive, non-hypermutated cells (as we did in Fig. 2). We also used the synthetic data sets to verify that learning the model on out-of-frame sequences only does not bias our inference results: the model learned from synthetic out-of-frame TRB sequences differed from the true model by only 0.4 bits, compared to 0.3 bits when learned on both in-frame and out-of-frame sequences.

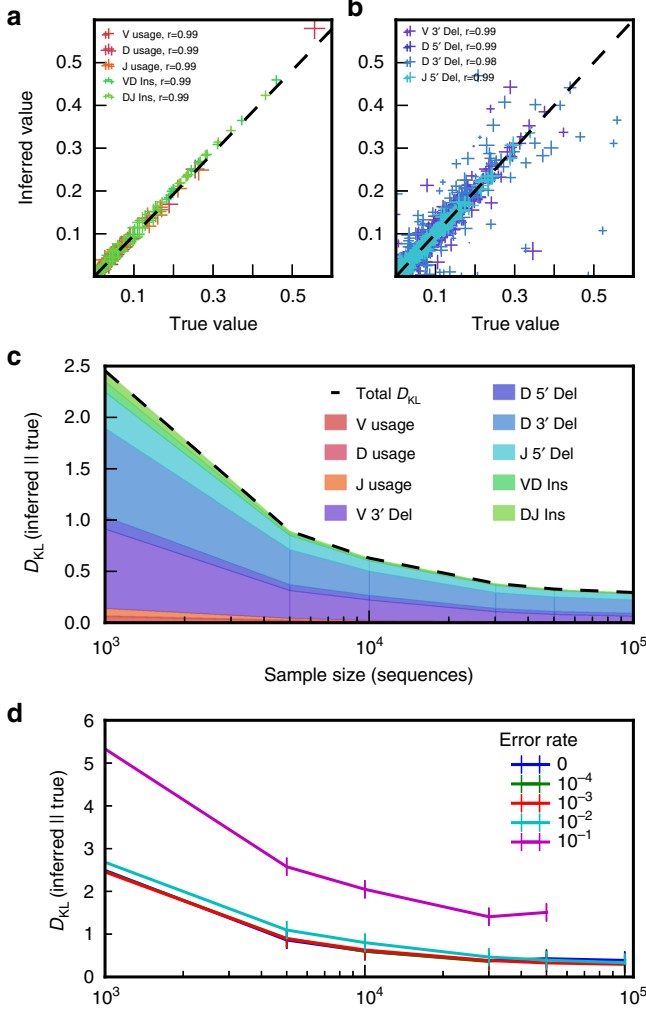

**Fig. 3** Validation on synthetic data. Short synthetic reads of recombined TRB sequences were generated with known recombination statistics, and given to IGoR as input to reinfer these statistics. Inference with $10^5$ TRB sequences and a typical sequencing error rate of $10^{-3}$ gives excellent agreement for **a** gene usage and insertion statistics and **b** deletion statistics (Pearson's $r$ for deletions is calculated on the joint statistics of gene usage and deletion number; cross size scales with gene usage). **c** Discrepancy between true and inferred values of the recombination statistics for TRB, measured by the Kullback–Leibler divergence, as a function of the number of unique sequences in the sample, and decomposed according to the features of the recombination scenario. **d** Same as **c**, for increasing rates of sequencing errors

**Analysis of scenario degeneracy**. By considering all recombination scenarios that contribute to the probability for each sequence, our approach departs significantly from most existing methods, whose goal is to find the most likely one (note however that Partis[10] treats scenarios probabilistically). To assess how often the most plausible scenario is the correct one, we analyzed synthetic sequences for which the generation scenario is known in a model without hypermutations. For each generated sequence, we used IGoR's analysis mode to enumerate the set of scenarios that were consistent with the nucleotide sequence, and ranked them according to their likelihood. Figure 4a shows the distribution of the rank of the true recombination scenario for TRB and unmutated IGH synthetic data. The maximum-likelihood scenario is not the correct one in 72% of 130 bp IGH sequences

and 85% of 60 bp TRB sequences. Both distributions have long tails, meaning that a substantial fraction of sequences has a very large recombination degeneracy. This degeneracy is not due to our inability to learn the correct model but it results from the inherent stochastic nature of the VDJ recombination process: different combinations of inserted and deleted nucleotides can produce exactly the same sequence. More data, longer reads, or a different model structure cannot improve this inherent limitation.

We then estimated how many scenarios, ranked from most likely to least likely, were needed to explain a given fraction $f$ of the total sequence likelihood. The distributions of this number for IGH across 100,000 generated sequences are shown in Fig. 4b for various values of $f$ (see Supplementary Fig. 10 for the equivalent plot for TRB data). To enumerate the correct scenario with $f = 95\%$ confidence requires to include at least 30–50 scenarios. This analysis indicates that, for both TCR and BCR, many scenarios need to be considered to correctly characterize the generation process.

IGoR outputs the probability of generation of the processed sequences, by summing the probabilities of all their possible scenarios, which deterministic assignment methods cannot do. It was shown that this generation probability was predictive of sharing properties between healthy individuals[12,15]. This functionality could be used as a useful indicator of convergent recombination in studies attempting to identify antigen-specific or auto-immune-related sequences from large clinical data sets.

**Comparison to other methods**. We compared our method to two representative state-of-the-art algorithms: MiXCR[8], an efficient assignment tool that finds the best matching germline genes, and Partis[10], a BCR-specific tool that uses maximum likelihood to find the most plausible scenario with a focus on calling hypermutations. 130-base-pair IGH hypermutation-free sequences were synthesized in silico from a data-inferred model using IGoR's generation mode. We then assigned recombination scenarios using MiXCR, Partis, and IGoR, and compared them to the true scenarios with which sequences were generated. In IGoR's and Partis' case, the model parameters were learned from the generated data set to mimic the analysis of real data. Figure 4c shows the performance of the three methods in assigning the correct scenario of recombination. IGoR performs about two times better than MiXCR and Partis in predicting the complete recombination scenario (in absence of hypermutations for IGH), as well as each of its individual components. Note that Partis does not include palindromic insertions, which both IGoR and MiXCR treat by appending a short palindromic sequence at the end of each germline segment; restricting the analysis to sequences generated without palindromic insertions makes Partis' performance comparable to that of MiXCR (Supplementary Fig. 11). Note that longer reads would improve the performance of all methods when assigning the V gene choice due to increased sequence information, but would not change the ability of other methods to identify the insertion or deletion profiles, or correctly call the real generation scenario. Since IGoR enumerates all likely scenarios, it is slightly slower than other methods (see Table 1).

To check that IGoR's performance is robust to changes in the sequence ensemble, and in particular to selection effects following recombination which are not modeled by IGoR, we generated an artificially selected naive IGH data set by selecting synthetic sequences according to their CDR3 length, so that the resulting length distribution exactly matched that of naive productive IGH sequences, using rejection sampling. Applying IGoR, Partis, and MiXCR to these sequences gave very similar performances as on unselected sequences, whether IGoR's model parameters were learned from unselected sequences (mirroring our approach of

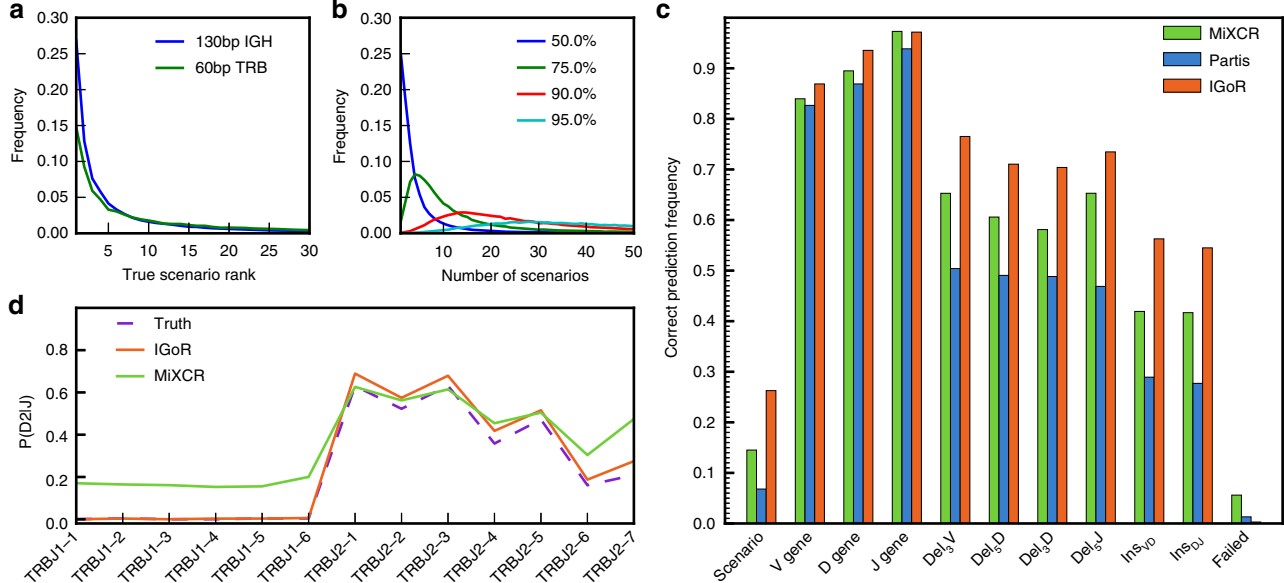

**Fig. 4** Probabilistic analysis of putative recombination scenarios and comparison to existing methods. Synthetic 130-bp reads of recombined hypermutation-free IGH sequences and 60-bp reads of TRB sequences were generated with a $5 \times 10^{-3}$ error rate, and processed for analysis by IGoR and two existing methods, MiXCR[8] and Partis[10]. IGoR ranks putative scenarios by descending order of likelihood. **a** Distribution of the rank of the true scenario as called by IGoR for both TRB and IGH. Note that the best-ranked (maximum-likelihood) scenario is the correct one in less than 30% of cases. **b** Distribution of the number of scenarios that need to be enumerated (from most to least likely) to include the true scenario with 50% (blue), 75% (green), 90% (red), or 95% (cyan) confidence for IGH (see Supplementary Fig. 10 for equivalent figure for TRB). **c** Frequency with which IGoR, MiXCR, and Partis call the correct scenario of recombination as the most likely one ("scenario") in hypermutation-free IGH, as well as each separate feature of the scenario ("V gene," etc.). "Failed" corresponds to sequences for which the algorithm did not output an assignment. **d** Usage frequency of TRB D gene conditioned on the J gene, inferred by the IGoR and MiXCR (Partis does not handle TCR sequences). IGoR recovers the physiological exclusion between D2 and J1, while MiXCR does not

### Table 1 Runtimes of the compared methods

| Chain | (Pre)alignments | | | Probabilistic treatment | | | |
|---|---|---|---|---|---|---|---|
| | MiXCR | Partis | IGoR/RepgenHMM | MiXCR | Partis | RepgenHMM | IGoR |
| TRA 100 bp | $2 \times 10^{-4}$ | NA | 0.3 | NA | NA | $10^{-2}$ | $10^{-4}$ |
| TRB 60 bp | $3 \times 10^{-4}$ | NA | 0.1 | NA | NA | 1 | 0.1 |
| IGH 130 bp | $10^{-3}$ | $10^{-2}$ | 0.2 | NA | $5 \times 10^{-2}$ | >1 | 0.2 |

Times for alignments are in seconds per sequence. In the probabilistic treatment, times are in seconds per sequence per iteration for Partis, and in seconds per sequence per EM iteration for IGoR and repgenHMM. MiXCR only performs alignments and does not include a probabilistic treatment. Partis only handles BCRs. IGoR and repgenHMM share the same code for pre-alignment. NA not applicable

learning the model from unproductive sequences), or from selected sequences (Supplementary Fig. 12). While IGoR's performance is robust in the annotation task, the inferred recombination statistics may still differ significantly between the selected and unselected sets[16,20].

Next, we compared the recombination statistics learned by the three methods to the true statistics used to generate the data. For MiXCR and Partis, we built the distribution of recombination events assigned to each sequence, while for IGoR these distributions were inferred using expectation-maximization, as explained before. All three methods yielded similar statistics for V and J gene usage and deletion profiles (see Supplementary Fig. 13 for IGH). However, the dependency between D and J usage in TRB (Fig. 4d) is correctly captured by IGoR but not by MiXCR (Partis was not included in this comparison as it does not handle TCR). TRB D and J genes are organized in two clusters, one containing D1 followed by genes of the J1 family, the other containing D2 followed by genes of the J2 family. Because of this organization, D2 cannot be recombined with genes from the J1

family[21]. MiXCR assigns 20% of impossible D2–J1 recombination events to sequences. By contrast, IGoR correcly learns the rule by assigning zero frequency to these impossible D–J pairs. The same results are obtained directly on real data (see Supplementary Fig. 14). Finally, IGoR accurately reconstructs the distribution of insertions, while the other methods systematically overestimate the probability of zero insertions (shown in Supplementary Fig. 13a, b for IGH).

IGoR also shares some features with repgenHMM[17], which represents scenarios probabilistically using a hidden Markov model. However, compared to IGoR, repgenHMM is much slower (Table 1), cannot call the most likely scenario(s), does not include a hypermutation model, and cannot encode arbitrary dependencies between recombination events. To illustrate this last point, we learned a model from synthetic TRA data simulated with an artificial dependency between V gene choice and number of insertions. While such dependency does not exist in TRA data, it may be relevant for other receptors (e.g., TCRδ and TCRκ) or for artificially designed receptors. Briefly, either one of two

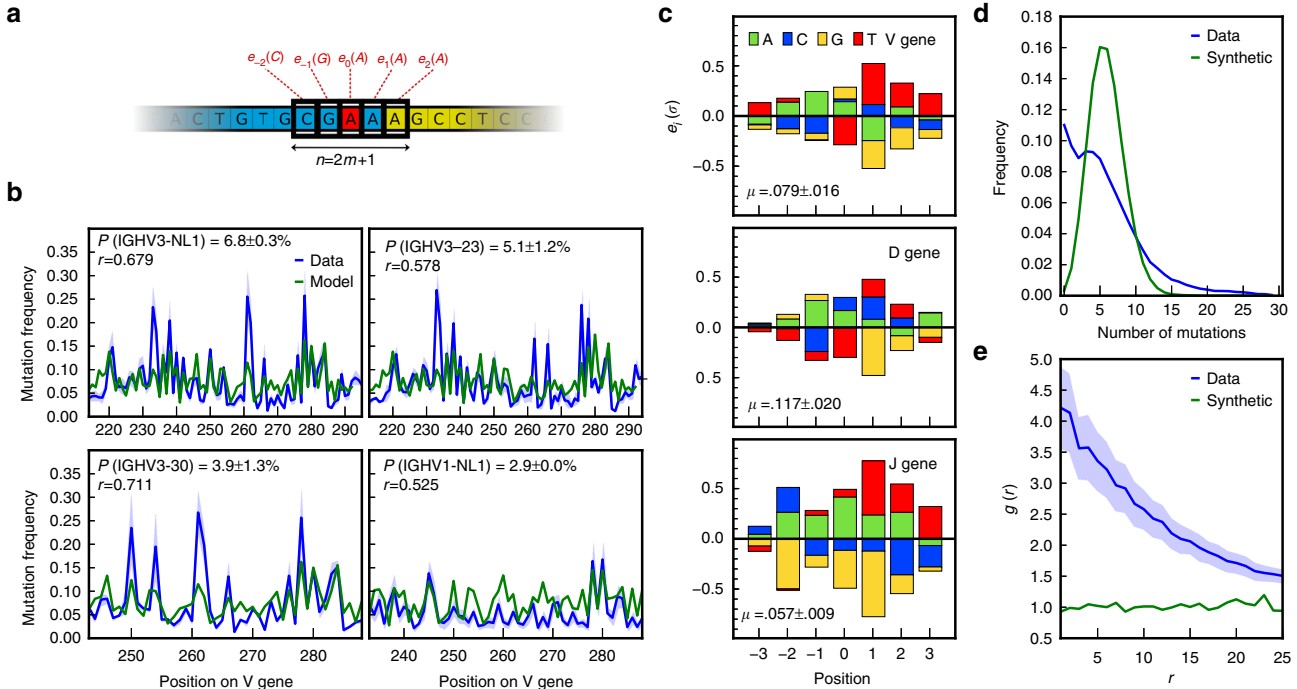

**Fig. 5** Hypermutation landscape. **a** Position weight matrix (PWM) model for predicting hypermutation hotspots in IGH. Each nucleotide $\sigma$ at position $i$ within $\pm m$ of the hypermutation site (in red) has an additive contribution $e_i(\sigma)$ to the hypermutation log odds (Eq. (3)). The PWM is learned by expectation-maximization from the out-of-frame sequences of memory B cells. **b** Comparison between the observed mutation rate per nucleotide and its prediction by the PWM model, as a function of position along the V segment, for the four most frequent V genes. Pearson correlation coefficient $\rho$ and gene usage are given for each. **c** PWMs inferred from the V, D, and J genes. **d** Distribution of the number of mutations in each sequence. Data sequences have a broader distribution than predicted by the model (as computed from generating synthetic sequences and mutations with a data-inferred 7-mer PWM model). **e** Spatial co-localization index $g(r)$, measuring the overrepresentation of pairs of hypermutations at genomic distance $r$ from each other. Synthetic sequences have $g(r) \approx 1$ by construction (green)

insertion profiles (the true one, or a geometric distribution) was assigned randomly to each V gene segment. In addition to learning the correct distributions of insertions for each V gene, which repgenHMM cannot by construction, IGoR learned the distribution of deletions, whose gene-dependent structure is the same in both methods, with much higher accuracy than repgenHMM (Supplementary Table 1).

**Somatic hypermutations**. To study patterns of SHMs in BCR expressed by memory B cells, we included into IGoR the possibility to infer a sequence-dependent hypermutation rate. The probability of error or mutation at a given position on the nucleotide sequence is assumed to depend on its immediate n-mer context (see Fig. 5a), through the logistic transformation of an additive score computed using a position weight matrix (PWM), similar to binding energy motifs used to describe DNA-binding sites[22]. We ran IGoR on memory out-of-frame IGH sequences from ref. [14] to learn 7-mer PWMs, as well as overall mutation rates (the geometric mean of the mutation rate over all possible 7-mers), while fixing the recombination statistics to those previously learned from naive sequences, using expectation-maximization (see "Methods" section). IGoR's probabilistic framework handles the degeneracy of sequence origin caused by convergent combinations of gene choices and hypermutations. The learning procedure differs crucially from ref. [16], where hypermutation rate was uniform. It also differs from Partis[10], which does not learn a PWM model but exhaustively infers the hypermutation profiles directly as a function of position for each gene (similarly to IGoR's posterior mutabilities of Fig. 5c, see below). For this reason, Partis learns a more complete

hypermutation profile, with more parameters. Other context-dependent models, with no additive assumption and hence large numbers of parameters learned from either silent mutations statistics[23] or from non-functional sequences in mutant mice[24], have also been proposed. Three distinct PWMs were learned for V, D, and J templated regions (Fig. 5b). To validate our PWM and mutation rate learning algorithm, we generated synthetic data with hypermutations according to the model learned from the real data set, and re-learned its parameters using IGoR, finding excellent agreement (Supplementary Fig. 15).

The PWM prediction for the position-dependent probability of hypermutations correlated well with that actually observed in the sequences ($r = 0.7$ for V genes, see Fig. 5c and Supplementary Fig. 16). PWMs were very reproducible across the two tested individuals ($r = 0.98$, Supplementary Fig. 17), indicating that the inference procedure is robust to the individual history of infections, and pointing to the universal nature of the SHM mechanism. By contrast, the inferred overall mutation rate differed by a twofold factor between the two individuals, probably owing to differences in age, past infections, or lifestyle (Supplementary Fig. 17). The motifs we found recapitulate previously reported hotspot motifs (positive values of the PWM) for every gene, including WRCY (or WRCH[25]) and WA[26,27] (W = A or T, Y = C or T, R = G or A; mutated position underlined), as well as cold-spot motifs albeit to a lesser extent (SYC, where S = C, G)[28]. In all three motifs, C and G are generally underrepresented, except for the mutated position in V and D genes, where T is less mutated than others. We assessed the robustness of the model to n-mer length by learning PWMs of sizes ranging from 3 to 9 (Supplementary Fig. 18). The contributions of each

relative position did not change substantially as a function of context length. Positions at least up to 4 nucleotides away from the mutation locus contribute to the motif. This could mean that the context dependence is broad, or alternatively that the motif model is indirectly capturing non-contextual effects. Overall, the inferred PWMs give both a more detailed and more nuanced view of the rules that govern hotspot positions, and cannot be reduced to a few easily describable motifs.

Figure 5b shows that the motifs differ substantially between V, D, and J genes. V-learned PWMs only moderately predict J gene hypermutation rates ($r = 0.5$ versus $r = 0.7$ for V gene rates), and J-learned PWMs predict V gene rates even worse ($r = 0.24$, see Supplementary Fig. 16). To assess whether this poor generalizability was due to assumption of additivity in the model, we learned a non-additive 5-mer model that tries to assign a specific hypermutation rate to each of the possible $4^5 = 1024$ possible 5-mer contexts (see Supplementary Note 2). In practice, we could only estimate mutabilities for a subset of 5-mer contexts comprising less than half (between 160 and 498) of all 1024 possible 5-mers. Models learned for the V and J segments were reproducible across individuals (Supplementary Fig. 19a–c), but did not agree well between V and J (Supplementary Fig. 19d, e). Although the model predicted the mutability very well as a function of position along the gene for the small subset of 5-mers for which that prediction was possible (Supplementary Fig. 20a, b), the model learned from V segments was not predictive of hypermutations rates in the J segment, and vice versa (Supplementary Fig. 20c, d), consistent with the results of the additive model. This disagreement indicates that predictions purely based on context-dependent motifs are insufficient to explain all of the variability in hypermutation probabilities, and that other mechanisms must be at play. The overall mutation rate was also different between germline genes, consistent with reports that the chromatin state affects hypermutation rates[29–31].

We then used the inferred PWM within IGoR to probabilistically call putative hypermutations in sequences. We first examined the distribution of the number of mutations in a sequence (Fig. 5d). The empirical distribution (red) is more skewed and has a longer tail than would be expected by assuming independent hypermutations in each sequence, as predicted by generating randomly hypermutated sequences with the inferred PWM (blue). This observation is consistent with the fact that different B cells have undergone a variable number of cycles of affinity maturation, resulting in differences in effective hypermutation rates. Second, we asked whether hypermutations co-localized within the same sequence, by calculating the enrichment of hypermutations at two positions as a function of their genomic distance (Fig. 5e). While this enrichment is one in synthetic sequences (since our model assumes that hypermutations are independent of each other), real data shows up to a fourfold enrichment of hypermutations at nearby positions. This difference is consistent with the fact that AID can cause repairs of DNA over large regions[32]. The typical distance at which the co-localization enrichment index decays gives an estimate for the length of these correlated regions of hypermutations, about 15 base pairs.

IGoR can in principle calculate the generation probability of any sequence. However, highly hypermutated sequences pose an additional challenge because the ancestral (unmutated) recombined sequence itself is sometimes not known with certainty. To overcome this issue, IGoR explores for each sequence all possible recombination and hypermutation scenarios, and calculates the generation probability of each potential ancestral sequence. Using synthetic data, we checked that the generation probability of individual sequences is well predicted by this method ($r = 0.97$, see Supplementary Fig. 21 and "Methods"

section), and its distribution accurately reproduced (see Supplementary Fig. 22).

## Discussion

By treating alignments of immune receptors to the germline probabilistically[15], IGoR corrects for systematic biases in the estimate of V(D)J recombination statistics, and predicts recombination scenarios more accurately than previous methods. Its detailed analysis of recombination scenarios further reveals that, even with a perfect estimator, the scenario is incorrectly called in more than 70% of sequences owing to the inherent stochasticity of the generation process, suggesting caution when interpreting results from deterministic assignments.

Although we demonstrated its functions on human TRA, TRB and IGH, IGoR's flexible structure makes it applicable to any variable lymphocyte receptor (TCR or immunoglobulin) and species for which a germline database is available. Unlike hidden Markov model-based methods (e.g., refs. [10,17]), it can include a wide array of possible dependencies between the recombination events. It can also be adapted to handle unusual or incomplete rearrangements (D–J rearrangments, DD2/DD3 rearrangements in TCR δ chains, hybrid TRA/TRD recombinations, etc.). IGoR can also help detect unusual rearrangement features by using its synthetically generated sequences as a control. For instance, rearrangements with tandem Ds have been reported[14], but distinguishing them from random insertions can be challenging. To test this, we counted sequences with two ≥10-nt D segments in the data, and compared it with predictions from IGoR's synthetic sequences generated with a single D segment (see "Methods" section). We found five times more double D assignments in IGH data than in the control, validating the findings of ref. [14]. In contrast, the same analysis performed on TRB showed no significant presence of tandem Ds. Future versions of IGoR should include the possibility of including multiple D segments . We also found that IGoR does not find reversed Ds in IGH (Supplementary Fig. 23).

IGoR infers recombination statistics from non-productive sequences only, but can do it with as few as 5000 sequences. Once a recombination model is learned for a given locus, IGoR can generate arbitrary numbers of synthetic sequences with the same statistics, which could be used as a control in disease-association studies, by helping to statistically distinguish antigen-specific clonotypes from public sequences with high convergent recombination frequencies, and thus dispense with the need of a healthy control cohort[33]. This approach is based on the high level of reproducibility of the receptor generation process across individuals[15], which allows one universal model to be used for different individuals. Gene usage profiles are the most personalized part of the distribution (in particular the V gene choice, which is expected to correlate with HLA type), but they contribute relatively little to the overall probability of generation of a receptor sequence[15,16]. To control for biases introduced by differential gene usages, it is possible to perform population analyses on specific V–J classes[34]. Alternatively, one could use IGoR to infer the V and J gene usage specific to the data set of interest while keeping the insertion and deletion profiles fixed, as those do not seem to depend on individuals or protocols.

Our analysis of hypermutations led us to infer distinct sequence motifs for mutation targets on the V, D, and J segments of human IGH, in contrast with previous approaches that assume a universal context model[23,24,27]. In this work, we focused our attention on additive models, because non-additive context models are limited by the number of n-mers for which the mutability can be estimated reliably. We checked that our results are not simply a consequence of this additive assumption, by

showing that they hold when learning a non-additive 5-mer model instead (Supplementary Fig. 19). Although the non-additive 5-mer model performed better than the additive model, it did so only on the few 5-mers for which hypermutation rates could be estimated reliably, which accounted for less than half of all 5-mers in the best of cases (Supplementary Fig. 20). In addition, any given 5-mer context is rare enough that it can often uniquely identify a gene segment and the position along that gene, making this better performance a likely result of overfitting.

We also compared our hypermutation rates to those of the "S5F" model[23] used for predicting hypermutation rates in human IGH. The agreement with our additive 5-mer model was quite poor (Pearson's $r$ ranging from 0.2 to 0.37, Supplementary Fig. 24). Correlation improved when comparing to our non-additive 5-mer model instead (Pearson's $r$ from 0.45 to 0.57, Supplementary Fig. 25), although that improvement was limited to the subset of well-sampled 5-mers. The predictability of the S5F model was comparable to that of the additive model (Supplementary Fig. 26). In particular, it predicted J segment mutabilities well, despite having been trained on V segments only. Note that the S5F model was trained on much longer reads and more diverse data sets than IGoR. Although our analysis of synthetic sequences showed that motifs can in principle be accurately learned from such short reads (Supplementary Fig. 15), applying IGoR to longer reads such as those with which the S5F model was trained could yield more robust models with better predictability.

We also found that hypermutations tend to co-localize along the sequence. Taken together, these results suggest that at least three effects determine hypermutation hotspots: the immediate DNA context of the hypermutation, as modeled by our sequence motifs, position-specific effects mediated by, e.g., chromatin configuration and histone modifications, and the co-occurence of nearby mutations. Future improvements of hypermutation target predictions should account for all three aspects, and rely on a better quantitative understanding of AID operation[31].

Apart from point mutations, the hypermutation process can also include insertions or deletions (indels)[35]. We estimate that 5–12% of the memory IGH sequences labeled as unproductive (i.e., with a frameshift or stop codon in their CDR3) also had an indel in their V region (Supplementary Fig. 27). IGoR currently discards about 30% of those sequences, owing to a large gap penalty that pushes those sequences below the minimum likelihood threshold. Although IGoR does not explicitly model indels in its probabilistic framework, we have checked that they did not affect our results: increasing the likelihood threshold to get rid of all sequences with indels does not affect the output of the inference. Including indels into the IGoR model structure would help better analyze repertoires that have higher indel rates due to enhanced hypermutation rates, as for example in untreated HIV carriers.

IGoR characterizes the elements of the VDJ recombination process and gives the overall probability of generating a given TCR or BCR receptor sequence. In order to characterize the statistics of the process—gene usage, insertion, and deletions profiles etc.—the model needs to be learned on sequences that have not undergone any kind of selection. Here, we have used out-of-frame sequences as an example of such sequences, but correctly sorted double-negative TCRs or pro-B cells could also be used when available. Trained in this way on selection-free sequences, IGoR's analysis module can then also be applied to functionally selected sequences to annotate them, and to compute their generation probability. Since selection only affects the sequence itself, and not directly its scenario, IGoR should perform as well on selected as on unselected sequences (see Supplementary Fig. 12). In addition, the sequence generation probability

computed by IGoR can be used to disambiguate the effects of generation from selection and identify sequences that have been selected for functional reasons. IGoR's model can also be trained directly on in-frame sequences from a given repertoire, even if this repertoire has undergone some form of functional selection (Supplementary Fig. 12). While the model structure may not be adapted to describe the various selective pressures acting on the translated amino acid sequence, such a model would still capture a combination of the sequence generation and selection forces, and could be useful for estimating the prevalence of particular sequences in selected repertoires. In general, as with all tools, it is important to understand the limitations of the data and the acquisition and pre-processing steps, which influence the interpretation of the results. IGoR can be used with data generated on any platform, with different experimental preparations, and its performance will not depend on the experimental technique as long as the data is trustworthy and correctly preprocessed.

Additionally to reads from repertoire sequencing, in the current version IGoR requires a germline database as input. Such databases are often incomplete as they do not include all possible polymorphisms across a population. In principle, highly recurrent "errors" identified by IGoR or other software could be used to detect polymorphisms and infer missing information from largely incomplete databases[36], extending the applicability of RepSeq sequencing to less well-characterized species. This approach would require enough knowledge of the genome to be able to construct primers and the experiment without knowing the full genome. It remains an interesting future application of IGoR.

In summary, since certain sequences are more likely to be generated than others, IGoR provides a baseline for how surprised we should be to see a given sequence in a given repertoire. It can be used as a tool for distinguishing convergent recombination from functionally selected sequences[33]. Finally, it can be used as a way to generate a control for studying affinity maturation, and can be combined with more accurate mutation models.

## Methods

**Overview of IGoR**. IGoR functions according to three modes: VDJ statistics learning, sequence analysis, and sequence generation. All modes rely on an explicit stochastic description of the recombination and hypermutation events. In the analysis and learning modes, each sequence is analyzed by listing all possible recombination and hypermutation scenarios. The learning mode iterates the analysis mode by updating the model parameters according to an expectation-maximization algorithm.

**Recombination model**. In all three modes, IGoR assumes that receptor sequences result from a recombination scenario comprising several stochastic elements—choice of germline segments, deletions, and insertions. These features are stochastic and share statistical dependencies with each other. For tractability, we assume that these dependencies can be represented by an acyclic graph, also called Bayesian network (see Supplementary Note 1 for details). This structure can be configured within IGoR's setup files. For the purpose of this study, we used the following dependency structures for the $\alpha$ chain of T cells (TRA):

$$P^{\alpha}_{\text{recomb}} = P(V, J)P(\text{del}V|V)P(\text{del}J|J)$$
$$\times P(\text{insVJ}) \prod_{i}^{\text{insVJ}} P_{\text{VJ}}(n_i|n_{i-1}), \qquad (1)$$

and for the $\beta$ chain of T cell receptors (TRB) and heavy chain of B cell receptors (IGH)[17]:

$$P^{\beta/H}_{\text{recomb}} = P(V, D, J)P(\text{del}V|V)$$
$$\times P(\text{insVD})P(\text{del}Dl, \text{del}Dr|D)$$
$$\times P(\text{insDJ})P(\text{del}J|J)$$
$$\times \prod_{i}^{\text{insVD}} P_{\text{VD}}(n_i|n_{i-1}) \prod_{i}^{\text{insDJ}} P_{\text{DJ}}(m_i|m_{i-1}), \qquad (2)$$

where $V$, $D$, and $J$ denote the choice of germline genes, del$V$, del$J$ the number of deleted base pairs at the ends of the V and J segments, delDl, delDr the number of

deletions at the left and right ends of the D segments, insVJ, insVD, insDJ, the numbers of insertions at each of the insertion sites (between V–J, or V–D and D–J), and $n_i$, $m_i$ the identities of the inserted base pairs. In the case of TRB, gene usage is further factorized as $P(V, D, J) = P(V)P(D, J)$.

**Context-dependent hypermutation model**. When processing TCRs or naive BCRs, a constant error probability is assumed throughout the sequence. When processing memory BCRs, a context-dependent hypermutation model is assumed: at each position along the V, D, and J genes, a hypermutation occurs with probability $P_{mut}$, with

$$\frac{P_{mut}}{1 - P_{mut}} = \mu \exp\left(\sum_{i=-m}^{m} e_i(\pi_i)\right), \tag{3}$$

where $(\pi_{-m}, \ldots, \pi_m)$ is the $(2m + 1)$-mer sequence context centered around the location of the mutation. The entries of the position weight matrix (PWM), $e_i(\pi)$, contribute additively to the motif, and $\mu$ is the overall hypermutation rate.

**Alignment to germline and scenario listing**. In the analysis and learning modes, each sequence is first aligned to all possible germline genes retrieved from germline databases (e.g., IMGT), using the Smith–Waterman algorithm[37]. Only germline genes with alignment scores higher than an adjustable threshold are considered for further analysis (see Supplementary Note 5 for details). Possible scenarios are then listed by picking germline genes with an above-threshold alignment score, and by choosing a number of base pairs to further delete from the ends of their aligned parts. The base pairs located between the germline segments trimmed in this manner are called insertions, and alignment mismatches to the germline are called errors or hypermutations. When the palindromic end of germline genes is not entirely deleted, the number of remaining palindromic base pairs are described as negative deletions. To allow for the possibility that the D segments be inserted in both directions in BCRs, we added the reverse complements of each D germline segment to the list of germline templates.

**Sequence analysis**. For each sequence in the data set, the probability of possible scenarios is computed using the recombination probability of Eq. (1) or (2), multiplied by the probability of errors or hypermutations $P_{err}$: $P_{scenario} = P_{recomb} \times P_{err}$. Scenarios are then listed in order of decreasing probability. The sum of probabilities $P_{recomb} \times P_{err}$ of possible recombination and hypermutation events gives the probability of observation of that particular sequence read, $P_{read}$. The probability that the pre-mutation sequence was generated by recombination, $P_{gen}$, is defined as the sum of the probabilities $P_{recomb}$ of scenarios leading to that sequence. Since the pre-mutation sequence is not known with certainty, we calculated an approximate generation probability $P_{gen}$ as the geometric mean of $P_{gen}$ of all possible unmutated sequences consistent with the read, weighted by their posterior probabilities, $P_{gen} \times P_{err}/P_{read}$. Alternatively, we approximated $P_{gen}$ by that of the most likely pre-mutation sequence (see Supplementary Note 4).

To shorten computation times, only plausible scenarios are listed by IGoR. Scenarios are enumerated by exploring the nodes of a hierarchical decision tree, where each depth corresponds to the choice of a scenario feature. Branches of the tree are discarded if their total contribution to the sequence probability is upper bounded to be below a certain threshold. Details of the procedure are given in the Supplementary Note 5.

**Learning algorithm**. The learning algorithm infers the parameters of Eq. (1) or (2), as well as the error or hypermutation model parameters of Eq. (3), from a large data sets of unique sequences. It relies on the sequence analysis module, and follows an expectation-maximization procedure. Starting from an arbitrary (but reasonable) set of parameters, all sequences in the data set are analyzed as described above, producing a long list of scenarios associated with each sequence. We define the pseudo-log-likelihood as the weighted sum of the log-likelihoods of all scenarios of all sequences, where the weights are given by the conditional probabilities of scenarios given the sequence, $P_{recomb}/P_{read}$ (expectation step). This pseudo-log-likelihood is then maximized with respect to the parameters of the log-likelihoods (Eqs. (1), (2), and (3)), while keeping the weights fixed. The parameters are updated, and the procedure repeated, until convergence. Mathematical derivations of the update rules and details about expectation-maximization are given in the Supplementary Note 2.

**Validation of model inference**. To compare the model parameters $\theta_1$ inferred from synthetic data to the known model parameters $\theta_2$ from which these data were generated, we computed the Kullback–Leibler divergence between two probability distributions, $D(\theta_1 \parallel \theta_2) = \sum_E P(E, \theta_1)\log[P(E, \theta_1)/P(E, \theta_2)]$, where the sum is over all scenarios E. $P(E, \theta)$ is computed using Eq. (1) or (2). This Kullback–Leibler divergence can be decomposed into additive contributions from each of the scenario features, as detailed in the Supplementary Note 3.

**Correlations between hypermutations**. To evaluate correlations between the occurence of hypermutations at close-by positions along the BCR sequence, we

computed the radial disbribution function defined as:

$$g(r) = (1/N_r) \sum_{V;(i,j)\in C_V(r)} f(i, j, V)/f(i, V)f(j, V), \tag{4}$$

where $f(i, V)$ and $f(i, j, V)$ are the frequencies of hypermutations at position $i$, and at both positions $i$ and $j$, respectively, calculated from individual scenario statistics weighted by their posterior probabilities. $C_V(r)$ is the set of pairs of positions separated by $r$ that were observed a large enough number of times in gene $V$, and $N_r = \sum_V |C_V(r)|$.

**Usage of tandem D segments**. In order to assess the occurrence of double D insertions during the VDJ recombination event of IGH or TRB, we computed the frequency with which one could align (with the Smith–Waterman algorithm) two non-overlapping Ds over at least 10 nucleotides, between the best V and best J alignments. We then compared the frequency obtained for synthetically generated sequences, to that obtained for real sequencing data.

**Software availability**. IGoR along with example data sets and pre-learned human TRA, TRB, and IGH models is available at https://github.com/qmarcou/IGoR.

**Data availability**. We applied the learning algorithm on the following publicly available data sets: TCR $\alpha$ and $\beta$ chains RNA data sets from ref. [12] are available on sequence read archive (SRP078490); TCR $\beta$ chains 60-bp DNA data sets from ref. [15] are available at http://physics.princeton.edu/ccallan/TCRPaper/data/; naive and memory BCR heavy chains DNA data sets from refs. [16,38] are available at http://physics.princeton.edu/ccallan/BCRPaper/data/.

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

## Acknowledgements

The work was supported by grant ERCStG no. 306312.

## Author contributions

T.M. and A.W. designed the study. Q.M. wrote the software code, processed the data, and prepared the figures. Q.M., T.M. and A.W. analyzed the data. Q.M., T.M. and A.W. wrote the paper.
