## [Peer Review File · Nature Communications]

Reviewers' comments:

Reviewer #1 (Remarks to the Author):

Before we begin, I should note that I (Erick Matsen) am a co-author of the partis program which has some overlap in functionality with IGoR, and that my postdoc Duncan Ralph also contributed to this review. Thus although I have done my best to be objective and we maintain a friendly academic rapport with the authors, we are officially speaking "competitors" and so the review should be filtered through that lens. I hope the AE will indicate if any of these points seem invalid.

IGoR is the latest iteration of a very impressive and important body of work extending back to 2012 concerning probabilistic models for repertoire development. It pulls together many more recent developments that have been published from this group. The paper is for the most part clearly written and well organized.

Major points

1.

Upon seeing this paper, I think many readers will want to know how IGoR is different than the repgenHMM software by almost all of the same authors (<https://www.ncbi.nlm.nih.gov/pubmed/27153709>). This is described but could be strengthened. One of the main differences is that IGoR isn't actually a HMM, but has a more sophisticated probabilistic structure. However, in reading the repgenHMM paper, we find a clever work-around that allows for a non-trivial joint distribution between germline gene identities:

> In general however, the V and J gene choices may be correlated in their joint distribution $P(V, J)$, breaking the Markovian nature of the rearrangement statistics. To preserve the Markovian structure, we built a separate HMM for each choice of the pair of germline genes (V, J), and use the forward-backward algorithm to calculate the marginals of the other rearrangement events conditioned on that choice.

We read also that IGoR has a more sophisticated mutation model. This begs the question: what do the new methods bring to the table in terms of performance compared to repgenHMM? Thus I suggest the easiest possible validation exercise: just compare the performance of these two tools, developed in the same lab, to each other. Also note that there are a number of things that could be incorporated, such as indel support, that could really make this version stand out from repgenHMM, but are not done. In addition, any model differences with previous versions could be called out more explicitly.

2.

Indels happen in somatic hypermutation, as clearly shown in Yeap et al 2015 (<http://dx.doi.org/10.1016/j.cell.2015.10.042>). Interestingly we find in the code (https://bitbucket.org/qmarcou/igor/src/c39ffde501d73bb6b46c7b70a746114c4d1d78a4/igor_src/Genechoice.cpp?at=master&fileviewer=file-view-default#Genechoice.cpp-101) the following TODO:

```
//TODO take into account in-dels and construct them in the constructed sequences
```

They have been shown to be important in clonal families responding to HIV. By looking at the gap penalties used in the Smith-Waterman step, I'm guessing that IGoR discards sequences with indels. I think this is a serious limitation that should be remedied or noted explicitly.

3.

The manuscript states "IGoR infers recombination statistics from non-productive sequences only," which corresponds with the goal of inferring recombination statistics for the generative process. However, people annotating (i.e. inferring deletion lengths for specific sequences) BCR and TCR data are primarily interested in doing so on in-frame sequences. If IGoR is meant for analyzing and annotating non-productive sequences only that should be stated clearly. However, if one should train on non-productive sequences and then apply those parameters to productive

sequences, then the current validation framework does not apply to this use case. In particular, the selected set of mutations will have a quite different distribution (changing the accuracy assessment for the tested parameters) and the joint gene usage distribution will differ as well. Thus the process of simulating and validating under the same model does not reflect performance under this use case.

4.

The marquee claim for this method, that it performs 2.5 times better than previous methods, has a lot of caveats. Most importantly, identical models were used for inference and simulation, which is unfair to other programs who formulate models differently. The only way that this would be justifiable is if the IGoR model generated exactly the same distribution on sequences as that in nature, which of course is not true. It's not even obvious that the IGoR model is the best in every respect: for example, partis uses a per-site empirical distribution for mutations at germline-encoded sites. I would suspect that such an empirical distribution has better performance under cross-validation than the 7-mer model (compare 5c, which is not terribly impressive IMO). Also note that IGoR itself only gets the metric used to make this claim (complete prediction frequency) right 25% of the time. This means that any slight shift in model assumptions, whether they be more or less accurate biologically (or correspond to in vs out of frame, e.g.), will greatly degrade performance according to the metric of exact match to the IGoR simulations.

5.

The paper uses different germline gene sets for different methods (Supplementary page 10). In our experience this is a very significant problem-- one simply cannot make a meaningful comparison of V call accuracy without synchronizing germline sets. We have seen 60% to 90% V accuracy changes when changing between germline sets supplied by different programs. It's also worth mentioning that on 130 base pair reads centered on the CDR3, a very large number of different V genes are degenerate in the 30-80 bases that aren't truncated (i.e. about 80% of the V is removed), whereas SNPs differentiating different Vs within a family are pretty much evenly distributed along the full length. This emphasizes the importance of using a modern read length, rather than 130 bp, for validation.

6.

The IGoR software doesn't seem finished, and it is very user-unfriendly. A quality software implementation would seem important to justify a new paper, because as far as I can tell all of the models in IGoR (perhaps with the exception of a slightly different conditional dependency structure) have appeared in other papers.

Here is a likely first experience with IGoR for a user asking for online help:

```
» ./igor -h
terminate called after throwing an instance of 'std::invalid_argument'
what(): Unknown argument "-h"
[1] 4992 abort (core dumped) ./igor -h
»
```

Please have a look at <https://academic.oup.com/gigascience/article-lookup/doi/10.1186/2047-217X-2-15>, which makes basic suggestions for software command line interfaces. The following issues are easily fixed, but make the software much less usable than it could be.

- as far as I can tell, the only way in which IGoR can fail is with a segfault: typo in command line flags, file not there, all segfault. Exception handling works great in C++.

- there is no way to install igor in the usual linux sense; `make install` can be executed but is broken. Thus one can't even run igor from the root of the git repository: even `igor_src/igor -run_demo` segfaults.

- the manual is incomplete, in particular in the description of the output files, and documents many flags that are NOT FUNCTIONAL YET. This doesn't correspond with a publication and version 1.0. The manual describes Python code for IGoR parsing, but the only Python I could find in the repository was for jemalloc. This must be fixed if the authors want anyone to use this software.

- I made a pull request to clean up the git repository, which I hope will be accepted.

- The supplementary material (section 2) "IGoR is designed in a modular way so the user can define arbitrary model forms". Does the user need to modify IGoR code, or is there some model definition syntax? If the authors are going to propose that users modify the IGoR code, some API documentation is needed.

7.

How frequently does IGoR infer contribution of palindromic nucleotides? It would be interesting to compare to:

Jackson, K. J. L., Gaeta, B., Sewell, W., & Collins, A. M. (2004). Exonuclease activity and P nucleotide addition in the generation of the expressed immunoglobulin repertoire. *BMC Immunology*, 5, 19. <https://doi.org/10.1186/1471-2172-5-19>

Details

It would have been much easier to review this MS if lines were numbered.

Main text:

Abstract:

- suggest "B or T-cell receptor sequence reads" rather than "B or T-cell receptors sequence reads"

P2:

- It would be nice to compare run times to other methods. Also, while it's certainly interesting to see the full run time distribution for different repertoires, the simple mean/total run time for one or two representative repertoires, directly compared to other methods, would I think be more interesting. Perhaps this could be added to the main text, and Fig S1 could remain in the supplement. It would also be nice to know the run times for full VDJ sequences, which are now much more common in research than the short ones used here.

- "In the case of TRB, gene usage is further factorized": why factorize only in the case of TRB?

- "BCR lights chains" should be "BCR light chains"

Fig1:

- I'm not quite sure how to interpret the diagram in Fig1b. In the text it's described as a Bayesian network, aka graphical model, but in that formalism I don't know what gray vs. black would mean. In addition, arrows mean conditional dependence, which doesn't seem to match up with the equation below the diagram. For example, the equation below the diagram shows a joint distribution of D and J, where the diagram shows there being a J distribution conditioned on D. Perhaps I nit-pick but I think it will be confusing for people accustomed to looking at graphical models.

- "segments in the case of IGH" -> "segments in the case of TCRB and IGH"

" each segments" -> "each segment"; and weighs -> weights

Fig2:

- I suggest specifying that you mean reproducible between `_individuals_`. If people think you mean reproducible between H/L/A/B they may disagree...

P4:

- At least mention that not all out of frame sequences are selection-free for their whole lifetimes? Indels can take an selected sequence out of frame.

- It's really interesting, and a little depressing, that gene usage varies more by sequencing protocol than individual (no change needed!).

P5:

- "IGH have significantly more insertions at the junctions than TCR": perhaps "IGH sequences"?

P6:

- "Can reach 1-10%" I'm not sure if you're referring to mean SHM rates, or the highest rates in a sample, but this seems low in either case. The mean rate across unsorted samples is perhaps 5-10%, and in our experience (which may be skewed by looking at African samples) the distribution is such that perhaps 5% or sequences have greater than 30% SHM.

- "By considering all possible recombination scenarios for each sequence, our approach departs significantly from most existing methods, whose goal is to find the most likely one." While some do not, most methods report, at minimum, a list of less-likely V/D/J gene assignments. Partis integrates over a number of recombination scenarios when doing clonal family inference. We can also contrast this "all possible scenarios" statement with the statement elsewhere in the manuscript: "Since exploring all possible scenarios would be computationally too costly, IGoR restricts its exploration to the reasonably likely ones."

P8:

- As mentioned above, 130 base pairs is extremely short by current standards -- assuming this includes the cdr3, it excludes most of the V. Why were such unrepresentative sequences chosen? I'd be surprised if this did not change the analysis of scenario degeneracy.

- " D an J": typo

- "usage in TRB is correctly captured by IGoR but not by the other methods": should read "by MiXCR" because partis was not included in this comparison.

Fig5:

- log oddS

P10:

- "lesser extend": the authors mean "lesser extent"

P11:

- "be reduced to a few easily describable motifs": indeed. The authors are certainly well aware that the field has moved on to more detailed descriptions, such as Yaari...Kleinstein 2013, but other papers before that from the Kleinstein lab were estimating such mutability rates. Cui...Kleinstein 2016 is cited later, but this part doesn't seem to do the literature justice.

P12:

- "synthetically": typo

Supplementary text:

Title: "highthrouput" typo

P1: "and the transition matrices": it's trivial, but I must ask why n's vor VJ and VD, but m's for DJ?

P2:

- latex thing: sometimes it's insVD, sometimes ins\$DJ\$

- "(in the following we often write S for S^(E) for brevity)": it's a little confusing then when they both appear in (4)

P5: typo in (18)

P6:

- "L the number of error-prone base pairs": they aren't just error "prone", are they?

P11: Leafs -> leaves

P7:

- Is there some approximation going on from (23) -> (24)? It appears that $\log(1+\exp(x))$ is being

replaced by x

Fig S7: "software one" -> "software on"

P9: that sequence is not known with certainty

Reviewer #2 (Remarks to the Author):

The paper describes IGoR, a tool for high-throughput immune repertoire analysis, and gives examples of its usage in several high throughput immune sequencing datasets. IGoR is built on previous publications of this group and makes the statistical framework developed in the past by them more accessible.

It is a well written paper describing a powerful statistical framework. I believe that the community will benefit from a detailed description of this tool, and therefore think that it should be published. Several comments are:

Throughout the paper, the data that the model is applied to is non-functional out of frame sequences, that are not subject to antigenic selection pressure. Few points are delicate here and require more explanation in the paper:

1. Could it be that these non-functional sequences are biased compared to functional sequences even after taking into account antigenic selection? That is to say, maybe, for example, the insertion distribution is enriched for numbers that cause the BCR/TCR sequence to be out of frame?
2. In B cells, it is important not to count multiple times sequences coming from the same clone as independently derived sequences. For this a clustering step is commonly proceeds the analysis and a represented sequence is taken from each clone. It is important to take this into account and to mention the crucial points in pre-processing the data.

There is a danger in applying such tool to the data in a "blinded" way. Cell population, sequencing technologies, and pre-processing, among other things, are crucial for the correct interpretation of the model. Applying it to pro-B cells could be very different from applying it to in- or out- of frame naive or plasma cell sequences.

It seems like the mutation model systematically underestimate mutation hot spots. Could the authors provide an explanation for this?

Minor comments/typos

- Abstract: "and its modular structure can investigate models" - reword. It reads as if the modular structure is the researcher.

- Page 3 Figure 1 caption: "Each segments gets trimmed a" - "segments" should be "segment"

- Figure 2: The deletion and insertion distributions are plotted but it is not clear for which V/J gene. Is it combined for all genes? How does it vary between genes?

- Page 4 Figure 2 caption: "s (b) V and (c) J genes." - switch V and (b) and J and (c).

- Page 6: "We found that the inference was highly accurate for datasets of 105 sequences and an error rate set to its typical experimental value, 10^{-3} (Fig. 3A and b), and was not affected by overfitting." - can the authors supply evidence that this was not affected by overfitting?

Reviewer #3 (Remarks to the Author):

In this manuscript, Marcou et al. present "IGoR", a novel tool for estimating and characterizing features of expressed antibody/B cell receptor and T cell receptor repertoires from high-throughput sequencing data. The authors present statistical assessments of their tool using simulated and real data (in both gDNA and RNA contexts), also offering direct comparisons to two additional existing tools. Although I believe the manuscript and analyses are technically sound and reach the bar for publication in Nature Comm., I would not be willing to recommend it for publication in Nature Communications in its present form. However, I believe just minor edits are required that provide more thoughtful background for motivating the need for such tools, as well as more thorough descriptions of their analyses and results in the context of the points raised in their discussion, so that the reader can better understand what is being presented. Please find my comments/questions below.

General Comments

1) In general, I found the level of detail, the layout of particular sections, and overall presentation of the results to be a little disjointed at times, jumping from one analysis to the next. For example, while I understand that IGoR is designed for use in both BCR and TCR datasets, at times it is not always clear to the reader when a distinction is made between deciding to present data on TCR vs. BCR datasets, or if both types of datasets were used to test all features of the software (excluding SHM). And in some cases when both TCR and BCR data are tested and presented on, this is done in different ways (i.e., using different plots between the two types of repertoire datasets, or only showing results from 1 of them?; e.g., Figures S2 and S3, or comparison of 130bp IGH vs. 60 bp TRB?). Further to this point, many of the supplementary figure legends and results sections pertaining to them are sparse in detail, to say the least, at times lacking even axis labels (e.g., Fig. S17). This makes take home messages difficult to understand at times. More thoughtful attempt to link the motivation for this work to the results, and in turn to more detailed discussion of its potential application would go a long way to improving the clarity and impact of this manuscript.

Minor Comments

1) The term "scenario" is introduced in the introduction section, but this should be explained in more detail to the reader, especially given it is one of the metrics used to assess performance, and it mentioned many times throughout the manuscript.

2) The "recombination of genomic segments" is not necessarily a "stochastic" process. This should be clarified.

3) Fig. 1a uses IGH as an example, whereas Fig. 1b uses TCR. Not sure whether this is intentional. In some ways it speaks to the general comment on clarity above.

4) Several abbreviations used throughout are not defined; e.g., "V", "D", "J", and "IGH".

5) "By contrast, V and J gene usage varied moderately but significantly across individuals, and even more across sequencing technologies, suggesting possible primer-dependent biases (Fig. S4, see also Fig. S17 for IGH D-J gene usage)." This is an interesting observation, though not surprising given previous studies. Did you test whether this variation influences IGoR performance. In other words, are certain libraries/techniques/individual repertoires better suited for IGoR? If so, why might this be?

6) "The maximum-likelihood scenario is not the correct one in 72% of IGH sequences and 85% of 60bp TRB sequences."

Should this read "72% of 130 bp IGH sequences...?"

7) "For an error rate of 10^{-3} , ~ 5000 unique out-of-frame sequences (which can be obtained from less than 2ml of blood with current mRNA sequencing technologies [14]) were sufficient to learn an accurate model of TRB (Fig. 3c)..."

Did you not test this explicitly for BCR? The sentences following the one quoted above suggest you have (as does Figure 3, S5), but again the results are presented in a confusing way.

8) Why were the read lengths used for TCR and BCR datasets selected? Does read length influence the performance of IGoR? Did you test? If not, some rationale for choosing the read lengths used seems warranted.

9) In Figure S8, from your comparison of IGoR to MiXCR and Partis you state: "Gene usage is mostly consistent between methods."

This doesn't seem to be necessarily true? For many IGHV genes, the data presented seems to indicate a good degree of variability between software packages (e.g., IGHV3-21 and IGHV3-30). Do the authors believe this is not relevant? If not, why?

Were IGoR and MiXCR at least compared for TCR analysis, even though Partis can't be used for TCR?

10) You mention the genomic data is a requisite for IGoR, and even comment that IGoR can be applied to any species for which genomic data is available. But what about species for which genomic data are available but germline databases are largely incomplete? Would you expect IGoR to perform less well in certain species, strains, or populations?

11) "Once a recombination model is learned for a given locus, IGoR can generate arbitrary numbers of synthetic sequences with the same statistics...". Is this referring to models for a locus within an individual, or more generally, in a species or population? One certainly seems more feasible than another based on what we understand from existing data. And it seems too early to make this point in such a definitive way, especially primarily based on the data presented here.

12) How do you reconcile these two statements in the discussion?:

"and thus dispense with the need of a healthy control cohort",

"Its detailed analysis of recombination scenarios further reveals that, even with a perfect estimator, the scenario is incorrectly called in more than 70% of sequences"???

These seem to conflict with one another.

Summary of changes

Reviewer #1 (Remarks to the Author):

Before we begin, I should note that I (Erick Matsen) am a co-author of the partis program which has some overlap in functionality with IGoR, and that my postdoc Duncan Ralph also contributed to this review. Thus although I have done my best to be objective and we maintain a friendly academic rapport with the authors, we are officially speaking “competitors” and so the review should be filtered through that lens. I hope the AE will indicate if any of these points seem invalid.

IGoR is the latest iteration of a very impressive and important body of work extending back to 2012 concerning probabilistic models for repertoire development. It pulls together many more recent developments that have been published from this group. The paper is for the most part clearly written and well organized.

Dear Erick and Duncan,

thank you very much for your thoughtful comments, pointers and positive evaluation of our work. We have added new analysis figures and text with which we hope we have addressed all your concerns. We have marked all the changes to the manuscript in red, except for corrected typos.

Major points

1.

Upon seeing this paper, I think many readers will want to know how IGoR is different than the repgenHMM software by almost all of the same authors (<https://www.ncbi.nlm.nih.gov/pubmed/27153709>). This is described but could be strengthened. One of the main differences is that IGoR isn't actually a HMM, but has a more sophisticated probabilistic structure. However, in reading the repgenHMM paper, we find a clever work-around that allows for a non-trivial joint distribution between germline gene identities:

> In general however, the V and J gene choices may be correlated in their joint distribution $P(V, J)$, breaking the Markovian nature of the rearrangement statistics. To preserve the Markovian structure, we built a separate HMM for each choice of the pair of germline genes (V, J) , and use the forward-backward algorithm to calculate the marginals of the other rearrangement events conditioned on that choice.

We read also that IGoR has a more sophisticated mutation model. This begs the question: what do the new methods bring to the table in terms of performance compared to repgenHMM? Thus I suggest the easiest possible validation exercise: just compare the performance of these two tools, developed in the same lab, to each other. Also note that there are a number of things that could be incorporated, such as indel support, that could really make this version stand out from repgenHMM, but are not done. In addition, any model differences with previous versions could be called out more explicitly.

Thanks for the suggestion. In general, IGoR can treat a model of arbitrary complexity given the elements of the VDJ recombination process, while repgenHMM exploits, and is therefore limited by, the Markov structure. But as you point out, you can “hack” repgenHMM in certain cases. Nevertheless, this work-around has some limitations. Before we illustrate these limitations in detail below, let us also note the following:

- RepgenHMM is not designed to analyze individual sequences and call scenarios. IGoR by contrast can give a list of most likely scenarios for each sequence.

- RepgenHMM is not designed to look at hypermutations, so it cannot learn a site-specific hypermutation model. Since it does not analyse scenarios for individual sequences as mentioned above, it cannot call hypermutations sites either.

We have done three types of additional validation exercises to illustrate these points and show how IGoR can handle datasets that repgenHMM cannot:

a. While the proposed work-around in repgenHMM is a fairly good solution for alpha and light chains, it is much slower for beta and heavy chains. We now compare runtimes of the two methods. IGoR runs much faster than repgenHMM on all datasets analysed, including on alpha chains for which it was designed. These results are now presented in a main text figure, along with comparison with other software (new Table I).

b. we simulated a model with a dependence between insertions and V gene choice and show in an additional Supplementary Table (Table S1) that repgenHMM is not able to correctly learn this model, while IGoR is. While this may seem like an artificial problem from the point of view of light and heavy or alpha and beta chains, delta and gamma chains are yet largely unexplored and there is evidence of more complicated generation patterns. Additionally, we know of groups that are working on using the VDJ recombination system to synthetically design receptors and barcodes. In this cases they often “program” in such artificial dependencies, so from our own experience it is useful to include this generalization.

c. repgenHMM does not learn the hypermutation model. It does learn an error rate that could be interpreted as constant hypermutation rate. IGoR goes further and uses the hypermutation model to refine the estimates of the generation process. We tried to show how this affects the learning procedure explicitly by simulating a heavy chain BCR model with different rates of hypermutations and then seeing if repgenHMM can correctly estimate the error rate as well as the elements of the recombination process, such as deletion profiles for high error rates. When we tried to learn the model, after 4 days of computation with ~15000 sequences repgenHMM had still not completed a single iteration. This is mainly because repgenHMM was not designed to handle high mutation rates. The figure we were hoping to make is not feasible since we do not have a model for light chains and doing hypermutations on TCR did not seem to make much sense to us. So this experience just reinforces point a of this response.

In summary, we have now added a discussion of the similarities and differences. IGoR arose from our own need to have a more powerful and fuller software to analyze datasets repgenHMM cannot handle, and go further in the analysis.

2.

Indels happen in somatic hypermutation, as clearly shown in Yeap et al 2015 (<http://dx.doi.org/10.1016/j.cell.2015.10.042>) . Interestingly we find in the code (https://bitbucket.org/qmarcou/igor/src/c39ffde501d73bb6b46c7b70a746114c4d1d78a4/igor_src/Genechoice.cpp?at=master&fileviewer=file-view-default#Genechoice.cpp-101) the following TODO:

```
//TODO take into account in-dels and construct them in the constructed sequences  
They have been shown to be important in clonal families responding to HIV. By looking at the gap penalties used in the Smith-Waterman step, I'm guessing that IGoR discards sequences with indels. I think this is a serious limitation that should be
```

remedied or noted explicitly.

Thank you for point out the interesting question of hypermutation indels. IGoR actually does not discard all sequences with indels in the germline, but rather imposes a large gap penalty. We have re-run the alignment procedure with a lower gap penalty in the germline alignment procedure, and found that only $\sim 1/3$ of sequences with indels were discarded in our inference. Sequences with indels make up ~ 5 or 12% of all sequences from the memory pool depending on the individual (see plot below), so that 3 to 4 % of sequences only were discarded. See additional figure S28 below:

Yet, we wanted to check whether discarding those sequences would affect the results of the inference. To answer this question, we increased the likelihood threshold for keeping sequences, from 10^{-70} to 10^{-40} . Doing so gets rid of all sequences with indels (and also incidentally of 40% of sequences without them, see below).

We reasoned that if changing that threshold did not affect the results of the inference, that would imply that the inclusion or not of sequences with indels would not have an impact on the inferred model. Indeed, repeating the model inference

with sequences resulting from this higher threshold yielded indistinguishable results (data not shown).

We now discuss indels in more detail in the discussion, and added the first plot above as an SI figure to give a sense of the number of sequences with indels in our dataset. We agree that more thorough analyses of indels using probabilistic methods such as suggested by IGoR would be interesting future directions of research.

3.

The manuscript states “IGoR infers recombination statistics from non-productive sequences only,” which corresponds with the goal of inferring recombination statistics for the generative process. However, people annotating (i.e. inferring deletion lengths for specific sequences) BCR and TCR data are primarily interested in doing so on in-frame sequences. If IGoR is meant for analyzing and annotating non-productive sequences only that should be stated clearly. However, if one should train on non-productive sequences and then apply those parameters to productive sequences, then the current validation framework does not apply to this use case. In particular, the selected set of mutations will have a quite different distribution (changing the accuracy assessment for the tested parameters) and the joint gene usage distribution will differ as well. Thus the process of simulating and validating under the same model does not reflect performance under this use case.

We have now added a paragraph in the discussion clarifying the potential applications of IGoR in terms of in-frame and out-of-frame sequences. IGoR is designed to learn the probability of generating a given sequence. The mechanistic model that IGoR encodes describes the outcome of a process that results in a pre-thymic selection sequence. IGoR does not account for selection of any kind, neither thymic nor somatic. For this reason, learning the model on out-of-frame sequences (or other sets of sequences that are known not to have undergone selection, such as double negatives or pro-B cells) ensures that the learned probability distribution is correct. However, one can learn the model on in frame sequences, being fully aware that the model one is learning is not “correct” and does not reflect any mechanistic structure, meaning the insertion, deletion and gene usage profiles should not be interpreted in a mechanistic way, but rather as a complex mixture of recombination mechanisms followed by selection. Since selection acts at the amino-acid level on the sequence as a whole, rather on the elements of the recombination scenario, it is expected to induce subtle, high-order correlations between these elements. However, IGoR could in principle still infer a good approximation of the probability distribution of sequences from the productive repertoire.

But despite these limitations when discussing applying the *inference* procedure to productive sequences, nothing keeps IGoR from processing productive sequences in the *analysis* mode, either to call scenarios, hypermutations, or to estimate sequence generation probabilities. Then, comparing the probability of seeing a productive sequence (or a hypermutation) under the model learned on out of frame sequences, one can identify signatures of selection, and tease out the generation biases from other processes such as selection. However, we agree that IGoR does not produce a selection model. We believe we validated our model correctly, as we did so in the framework of the generation process (no selection), but we stress that this generation process can be used as a “null” for detecting, teasing out, and analyzing selection effects (both thymic selection for TCR, and affinity maturation for BCR).

We should also stress that since selection on in-frame sequences acts only on the sequence, and not on the scenario that generated it, the posterior probability of

scenarios given the sequence is not expected to be affected by selection. As a result, our IGoR's annotation should still be valid on in-frame, selected sequences, even though it was learned on out-of-frame, unselected sequences.

We have explicitly tested the applicability of IGoR to annotate selected sequences by assuming a simple data-driven selection model, which probabilistically selects synthetically generated sequences according to their CDR3 length, so that the distribution of that length in the selected repertoire matches the observed one in the actual in-frame repertoire (using importance sampling). Applying IGoR on these selected sequences yields similar performance than on unselected sequences (Fig S21). In addition, learning model parameters from the selected sequences using IGoR's inference module, and using these parameters to do scenario assignment on selected sequences, yields very similar performance. This shows both that

- using IGoR learned from out-of-frame sequences on in-frame sequences is legitimate;
- in relation to point 4 below, adding selection to the dataset does not affect performance, even though that selection step is not part of IGoR's model assumptions.

4.

The marquee claim for this method, that it performs 2.5 times better than previous methods, has a lot of caveats. Most importantly, identical models were used for inference and simulation, which is unfair to other programs who formulate models differently. The only way that this would be justifiable is if the IGoR model generated exactly the same distribution on sequences as that in nature, which of course is not true.

Let us first clarify any potential confusion about this statement: the "2.5 times better" (now changed to 2x after using the same set of germline genes for MiXCR, see below) refers to IGoR's ability to characterize the basic generation process and not to learn hypermutations: The comparison was performed on synthetic BCR sequences with very low mutation rates, following the model learned from naïve BCRs. We now clarify this in the text.

Concerning the fairness of the comparison, the model we simulate is a general mechanistic model for the generation process: pick a V, D and J, add insertions and deletions. Simulating it with any other software with the same dependencies between these mechanisms would generate statistically identical sequences - there is no ambiguity there. IGoR's structure allows for any statistical dependency between any of the elements of recombination. In a way, it contains almost all other possible model choices. In this particular comparison the generation model was chosen to mimic the true BCR beta chain statistics as close as we could. In Elhanati et al. 2015 the set of relevant dependencies was analysed in detail, and we used that set in this comparison for biological realism. Of course we could add more dependencies, either in the generation process or in the learning process (and we have in this revision, see Table S1), but apart from potential overfitting one does not expect any change in the performance of IGoR or of other software on a synthetic dataset generated that way. We have also added an artificial selection layer on BCR synthetic sequences and showed that IGoR's performance was not affected by it (Fig. S21, see above).

Again, let us stress that this model does not include hypermutations. We compare IGoR to Partis without hypermutations, clearly stating that is not what Partis was designed to do. We completely agree that for hypermutations, the choice of the model matters, but this discussion is not about the hypermutation model. We

apologize for the confusion and we hope this point is now clear. We have changed the text to give a fairer account of this comparison.

We now recognize that IGoR has an advantage in this comparison due to the fact that its model structure can encode more general statistical rules of VDJ recombination. We note however that this is the reason why we think IGoR is novel and brings something previous software did not.

It's not even obvious that the IGoR model the best in every respect: for example, partis uses a per-site empirical distribution for mutations at germline-encoded sites. I would suspect that such an empirical distribution has better performance under cross-validation than the 7-mer model (compare 5c, which is not terribly impressive IMO).

We completely agree that Partis is going to learn the hypermutation model more accurately because it learns it exhaustively with greater details and more parameters, whereas IGoR takes another route and attempts to parametrize the hypermutation process with a model. We have now stated this clearly in the manuscript.

Also note that IGoR itself only gets the metric used to make this claim (complete prediction frequency) right 25% of the time. This means that any slight shift in model assumptions, whether they be more or less accurate biologically (or correspond to in vs out of frame, e.g.), will greatly degrade performance according to the metric of exact match to the IGoR simulations.

We completely agree that 25% is not good. This is in fact one of the points of our results, which we feel has not been emphasized enough before: due to the probabilistic nature of the recombination process, it is impossible to deterministically state how a given sequence was generated. We perform the first systematic analysis of this failure, with a break-down of how well methods can call different recombination elements. We show that failure often comes from the insertions and deletions – as a simple example, deleting and inserting the same set of nucleotides results in the same sequence. The hardest part of the model to learn are the deletion profiles and these are not model dependent - they are learned exhaustively. We agree that changing the analysed set (productive vs. non-productive, different cell subsets, tissues, etc.) could change this performance. However, testing this would be difficult because we need to know the actual recombination scenario, which means that we can't test it on real sequences. This would require an explicit model of generation and selection that could be criticized in its own right. That said, we believe that accounting for the specific insertion and deletion profiles of the recombination process is the single most important step for improving scenario prediction, as we show in our comparison, and this is unlikely to be affected in major ways by selection.

Again, we emphasize that this point is made on the model without considering hypermutations. We have now tried to clarify this in the text.

5.

The paper uses different germline gene sets for different methods (Supplementary page 10). In our experience this is a very significant problem-- one simply cannot make a meaningful comparison of V call accuracy without synchronizing germline sets. We have seen 60% to 90% V accuracy changes when changing between germline sets supplied by different programs. It's also worth mentioning that on 130 base pair reads centered on the CDR3, a very large number of different V genes are

degenerate in the 30-80 bases that aren't truncated (i.e. about 80% of the V is removed), whereas SNPs differentiating different Vs within a family are pretty much evenly distributed along the full length. This emphasizes the importance of using a modern read length, rather than 130 bp, for validation.

We thank you for pointing this out, which led us to discover that the set of germline genes was different in the different methods, and does indeed matter. We have now redone the analysis with the same set of germline genes in all software and our conclusions remain mostly unchanged (the 2.5x improvement now becomes a 2x improvement). We initially opted for simplicity and used the germline sets that were shipped with each of the packages but this was clearly an oversight because our generation software used a particular germline set that was different. We completely recognize that now. We have updated all the figures and made it clear we are using the same germline genes.

We chose the 130bp read length to mimick the data we are currently working with. We think this comment comes from working with slightly different datasets: the majority of the data we have access to come from hiSeq machines (<https://www.illumina.com/systems/sequencing-platforms.html>), which has 8 lanes of 2x100bp reads (maximum read length is 2x150bp but 2x100bp are cheaper and more accessible), which after removing primers, barcodes and other auxiliary sequences results in about 130bp reads. We agree miSeq reads are longer, they can be up to 300bp, but miSeq offers lower output. Many of our collaborators continue to use hiSeq because it offers much higher throughput, especially for TCRs: e.g. Britanova et al used 2x100bp in their recent paper (Chudakov group), Emerson et al have 87bp reads (Robins group). We suspect you are looking at bnAb data which maybe has longer reads but we do not have access to these datasets, and the dataset from the Robins group we analysed also had short reads.

We completely agree that having longer reads allows for a better characterization of the V segment. This would in fact make much IGoR faster, as it would have to loop over much fewer V choices, decreasing run times. Since most failures come from insertion and deletion assignments, we do not expect longer reads and better V assignments to affect our main results.

6.

The IGoR software doesn't seem finished, and it is very user-unfriendly. A quality software implementation would seem important to justify a new paper, because as far as I can tell all of the models in IGoR (perhaps with the exception of a slightly different conditional dependency structure) have appeared in other papers.

Here is a likely first experience with IGoR for a user asking for online help:

```
» ./igor -h
```

```
terminate called after throwing an instance of 'std::invalid_argument'
```

```
what(): Unknown argument "-h"
```

```
[1] 4992 abort (core dumped) ./igor -h
```

```
»
```

Please have a look at <https://academic.oup.com/gigascience/article-lookup/doi/10.1186/2047-217X-2-15>, which makes basic suggestions for software command line interfaces. The following issues are easily fixed, but make the software much less usable than it could be.

We thank the reviewer for the helpful link. IgoR now implements -h and -help commands in order to access the program's manual. This manual is now also accessible after installation through the 'man igor' command.

Following the reviewer's suggestion we have implemented all applicable suggestions

in the provided link such that:

- IGoR will now print a message when no commands were supplied
- `igor -v` and `igor -version` print IGoR's version
- error and log messages are passed to `stderr`
- paths handling is now dealt through `configure` and `installation` (see below)

We hope these improvements will make IGoR user friendly.

- as far as I can tell, the only way in which IGoR can fail is with a segfault: typo in command line flags, file not there, all segfault. Exception handling works great in C++.

IGoR used to terminate by throwing a C++ standard library exception upon failure following the user's input. Such exceptions send a SIGABRT (core dumped) signal to the operating system and not a segmentation fault that refers to the termination of the program by the OS following violation of memory.

However we agree with the reviewer that such a termination could prevent from correctly integrating IgoR in a pipeline and/or worry the user. IGoR now exits by returning `EXIT_SUCCESS` or `EXIT_FAILURE`, two operating system independent flags for success or failure of the program. We thank the reviewer for this suggestion.

- there is no way to install igor in the usual linux sense; ``make install`` can be executed but is broken. Thus one can't even run `igor` from the root of the git repository: even ``igor_src/igor -run_demo`` segfaults.

IGoR ships with the libraries (jemalloc and gsl) it needs in order to make it easier for the user to use the program. This initially created issues for the correct integration within the autotools suite (a tool suite guaranteeing automatic configuration for a wide variety of operating systems/hardware). We have now solved these issues and the `'make install'` command will install IgoR's executable, data (provided models and demo) and manual (through the `man` command) such that IGoR can be used from any directory.

- the manual is incomplete, in particular in the description of the output files, and documents many flags that are NOT FUNCTIONAL YET. This doesn't correspond with a publication and version 1.0. The manual describes Python code for IGoR parsing, but the only Python I could find in the repository was for jemalloc. This must be fixed if the authors want anyone to use this software.

We apologize for this oversight. All the parts that were marked as not functional yet are now functional. All python scripts have been concatenated and organized as an installable python package in the `'pygor/'` folder.

- I made a pull request to clean up the git repository, which I hope will be accepted.

We thank the reviewer for his help to maintain the git repository. We have accepted and integrated the pull request.

- The supplementary material (section 2) "IGoR is designed in a modular way so the user can define arbitrary model forms". Does the user need to modify IGoR code, or is there some model definition syntax? If the authors are going to propose that users modify the IGoR code, some API documentation is needed.

The input file can be modified without modifying the IGoR code. We have now

written a better description of the input file syntax in the README. Alternatively, models can be defined through the C++ high level functions, in the custom section of the main.cpp file. A dedicated section in the README file explains briefly this approach.

Because by designing IGoR we have also aimed at designing an API through which researcher from the field can, on top of existing ones, test new model components (e.g the different hypermutation models we have tested in this article) we now provide a documentation in pdf and browsable html format in the /docs and /docs/html folders describing the C++ interface. This documentation was generated using the Doxygen software for ease of maintenance and will guarantee better traceability and transparency.

We are extremely grateful for these very useful suggestions. We have attempted to implement all to them as we agree these are very important points.

7.

How frequently does IGoR infer contribution of palindromic nucleotides? It would be interesting to compare to:

Jackson, K. J. L., Gaeta, B., Sewell, W., & Collins, A. M. (2004). Exonuclease activity and P nucleotide addition in the generation of the expressed immunoglobulin repertoire. *BMC Immunology*, 5, 19. <https://doi.org/10.1186/1471-2172-5-19>

We present the palindromic nucleotides as negative deletions in the plots in Fig. 2B and 2C. For clarity we now also mention this in the main text. Comparing to Jackson et al, we do not see the same patterns of palindromic nucleotides for either V, D and J genes, or for given classes. We summarized our results in additional Fig. S20, shown below.

The discrepancy is not surprising since Jackson et al note that most of the nucleotides they identify as exonuclease removal are not palindromic sequences but are likely to be newly inserted nucleotides by TdT that happen to be the same nucleotide as the deleted one. They claim that maybe even as little as 9 out of the 2899 junctional nucleotides considered are truly palindromic. Given the small numbers, they cannot report the statistics for the nucleotides they believe are truly palindromic. In general, we have much more data and a much more reliable method for calling palindromic sequences.

Details

It would have been much easier to review this MS if lines were numbered.

We have now numbered the lines.

Main text:

Abstract:

- suggest “B or T-cell receptor sequence reads” rather than “B or T-cell receptors sequence reads”

Thank you. We have corrected this.

P2:

- It would be nice to compare run times to other methods. Also, while it’s certainly interesting to see the full run time distribution for different repertoires, the simple mean/total run time for one or two representative repertoires, directly compared to other methods, would I think be more interesting. Perhaps this could be added to the main text, and Fig S1 could remain in the supplement. It would also be nice to know the run times for full VDJ sequences, which are now much more common in research than the short ones used here.

We have added a new table in the main text (Table I) comparing the mean runtime of the different methods.

Increasing the reads to have longer Vs would decrease the runtime of the method. Since IGoR has to sum over all likely scenarios, having longer reads makes it easier to discard V gene candidates and decreases the summation time over scenarios.

- “In the case of TRB, gene usage is further factorized”: why factorize only in the case of TRB?

We now have a full TRB model and it gives essentially the same results. There are differences in the deletion profiles (DKL of 1 to 2 bits depending on the junction), but almost no difference to other features of the model, such as the insertion profiles. We have included the full model in the IGoR package. The discrepancy between the treatment of TRB and other chains was for historical reasons: we first did the TRB and then we were interested in other statistics so looked at the full model. This has now been uniformized.

- “BCR lights chains” should be “BCR light chains”

Thank you. We have corrected this.

Fig1:

- I’m not quite sure how to interpret the diagram in Fig1b. In the text it’s described as a Bayesian network, aka graphical model, but in that formalism I don’t know what gray vs. black would mean. In addition, arrows mean conditional dependence, which doesn’t seem to match up with the equation below the diagram. For example, the equation below the diagram shows a joint distribution of D and J, where the diagram shows there being a J distribution conditioned on D. Perhaps I nit-pick but I think it will be confusing for people accustomed to looking at graphical models.

Thank you for noticing. We have corrected the equation to reflect the diagram.

- “segments in the case of IGH” -> “segments in the case of TCRB and IGH”

Thank you. We have corrected this.

“ each segments” -> “each segment”; and weighs -> weights

Thank you. We have corrected this.

Fig2:

- I suggest specifying that you mean reproducible between individuals. If people think you mean reproducible between H/L/A/B they may disagree...

We have clarified this.

P4:

- At least mention that not all out of frame sequences are selection-free for their whole lifetimes? Indels can take an selected sequence out of frame.

We have clarified this. However, biology suggests that sequences that are made out-of-frame by a indel should not be able to get out of germinal centers, and therefore should not be seen in samples collected from the blood, as you discussed in a recent blog post:

<https://b-t.cr/t/are-there-many-non-functional-b-cell-receptors-which-have-experienced-the-effect-of-selection/458>

The data are consistent with this, as we see no significant increase of out-of-frame sequences in the memory pool (which has hypermutation indels) relative to the naïve pool (which does not). There was even a decrease, from 12.7% (individual 2) and 13.5% (individual 1) in the naïve pool to 8.1% and 11.3% in the memory pool (productiveness called by MiXCR).

- It's really interesting, and a little depressing, that gene usage varies more by sequencing protocol than individual (no change needed!).

We agree. We attribute this to problems related to unaccounted primer biases in efficiency.

P5:

- “IGH have significantly more insertions at the junctions than TCR”: perhaps “IGH sequences”?

We have changed this.

P6:

- “Can reach 1-10%” I'm not sure if you're referring to mean SHM rates, or the highest rates in a sample, but this seems low in either case. The mean rate across unsorted samples is perhaps 5-10%, and in our experience (which may be skewed by looking at African samples) the distribution is such that perhaps 5% or sequences have greater than 30% SHM.

We agree, Fig. 5D shows exactly what you are saying - there is a long tail with about 5% of sequences having more than 30% SHM, but the mean of the distribution is 1-10%. We have now plotted a percentage histogram in the SI (Fig. S22).

- “By considering all possible recombination scenarios for each sequence, our approach departs significantly from most existing methods, whose goal is to find the most likely one.” While some do not, most methods report, at minimum, a list of less-likely V/D/J gene assignments. Partis integrates over a number of recombination scenarios when doing clonal family inference. We can also contrast this “all possible scenarios” statement with the statement elsewhere in the manuscript: “Since exploring all possible scenarios would be computationally too costly, IGoR restricts its exploration to the reasonably likely ones.”

We have modified the statement to “By considering all recombination scenarios that contribute to the probability for each sequence” and we have added a sentence stating that Partis also is probabilistic.

P8:

- As mentioned above, 130 base pairs is extremely short by current standards -- assuming this includes the cdr3, it excludes most of the V. Why were such unrepresentative sequences chosen? I’d be surprised if this did not change the analysis of scenario degeneracy.

As we have discussed above in detail in response to point 5, for us and our experimental collaborators these are representative, currently generated sequences and this is precisely why we chose this length. Longer reads will decrease the uncertainty about V gene choice but will not change anything else. We now clarify this at the beginning of the Results section.

- “D an J”: typo

Thank you. We have corrected this.

- “usage in TRB is correctly captured by IGoR but not by the other methods”: should read “by MiXCR” because partis was not included in this comparison.

Thank you. We have corrected this.

Fig5:

- log oddS

Thank you. We have corrected this.

P10:

- “lesser extend”: the authors mean “lesser extent”

Thank you. We have corrected this.

P11:

- “be reduced to a few easily describable motifs”: indeed. The authors are certainly well aware that the field has moved on to more detailed descriptions, such as Yaari...Kleinstein 2013, but other papers before that from the Kleinstein lab were estimating such mutability rates. Cui...Kleinstein 2016 is cited later, but this part doesn’t seem to do the literature justice.

Thank you for this comment. We have added a longer discussion to describe the current literature. We have added the equivalent of the S5F (synonymous 5-mer functional) plots based on our 5-mer model to the SI (Figs S23-S24):

We note however that only a few 100's of the 1024 possible 5-mers are found in

sufficient amounts to evaluate their mutabilities reliably, which was our main motivation for proposing an additive model. We also note that, since each 5-mer is rare, it is often sufficient to specify in large part the gene and position of the hypermutation. When that's the case, the 5-mer model behaves like a gene- and position-dependent model, and it is not clear what predictive power it has. We have also compared our results directly to the S5F model made available online, both for the full 5-mer model, and the additive model (Figs. S25-S27). We discuss more clearly that our results show that the hypermutation model seems to be context dependent and cannot be explained by a universal n-mer model. We have now extended this discussion and have added a plot showing that the discrepancy is not due to the additive nature of the model (Fig S24):

P12:

- “synthetically”: typo

Thank you. We have corrected this.

Supplementary text:

Title: “highthroughput” typo

Thank you. We have corrected this.

P1: “and the transition matrices”: it’s trivial, but I must ask why n’s vor VJ and VD, but m’s for DJ?

We needed two dummy variables for the case of heavy chains, so we picked m and n. For light chains we only needed one. We agree this is not the mostly natural choice, but it minimizes the introduction of cumbersome notations. We’re of course open to better suggestions.

P2:

- latex thing: sometimes it’s insVD, sometimes ins\$DJ\$

Thank you. We have corrected this.

- “(in the following we often write S for $\hat{S}(E)$ for brevity)”: it’s a little confusing then when they both appear in (4)

We now put $\hat{S}(E)$ everywhere.

P5: typo in (18)

Thank you. We have corrected this.

P6:

- “L the number of error-prone base pairs”: they aren’t just error “prone”, are they?

Thank you. We have rephrased.

P11: Leafs -> leaves

Thank you. We have corrected this.

P7:

- Is there some approximation going on from (23) \rightarrow (24)? It appears that $\log(1+\exp(x))$ is being replaced by x

No, there is no approximation. The sum over E is redefined in terms of a sum over π with a density of states. Then some algebra follows.

Fig S7: “software one” \rightarrow “software on”

Thank you. We have corrected this.

P9: that sequence is not known with certainty

Thank you. We have corrected this.

Reviewer #2 (Remarks to the Author):

The paper describes IGoR, a tool for high-throughput immune repertoire analysis, and gives examples of its usage in several high throughput immune sequencing datasets. IGoR is built on previous publications of this group and makes the statistical framework developed in the past by them more accessible.

It is a well written paper describing a powerful statistical framework. I believe that the community will benefit from a detailed description of this tool, and therefore think that it should be published. Several comments are:

We thank the reviewer for his/her positive opinion about our manuscript and his/her detailed comments. We answer them below in detail. All corrections to the manuscript are marked in red.

Throughout the paper, the data that the model is applied to is non-functional out of frame sequences, that are not subject to antigenic selection pressure. Few points are delicate here and require more explanation in the paper:

1. Could it be that these non-functional sequences are biased compared to functional sequences even after taking into account antigenic selection? That is to say, maybe, for example, the insertion distribution is enriched for numbers that cause the BCR/TCR sequence to be out of frame?

This is a very good point and we have now re-learned the model from out-of-frame sequences only to show that there is no bias introduced by learning the model on non-functional sequences (i.e. with a frameshift in the CDR3). Doing so yields a very small difference with the true generating model (0.4 bits). This error is consistent with just sampling error, as it is almost the same divergence one gets when using the same number of non-restricted sequences (both in-frame and out-of-frame), ~ 0.3 bits. We have previously tested this in Murugan et al 2012, but we agree with the reviewer this is an important point and we reproduce this test here for a fuller manuscript.

This new control is now included in the main text.

2. In B cells, it is important not to count multiple times sequences coming from the same clone as independently derived sequences. For this a clustering step is commonly proceeds the analysis and a represented sequence is taken from each clone. It is important to take this into account and to mention the crucial points in pre-processing the data.

Both for BCRs and TCRs preprocessing steps are crucial. We now state this clearly in the results section. The analyzed data were preprocessed, either by the data provider itself in the case of data from the Robins lab, or by MiXCR in the case of the data from the Chudakov lab.

There is a danger in applying such tool to the data in a “blinded” way. Cell population, sequencing technologies, and pre-processing, among other things, are crucial for the correct interpretation of the model. Applying it to pro-B cells could be very different from applying it to in- or out- of frame naive or plasma cell sequences.

We completely agree with the reviewer. We have added a paragraph to the discussion about the applicability of the tool incorporating the Reviewer’s remark.

It seems like the mutation model systematically underestimate mutation hot spots. Could the authors provide an explanation for this?

One of the findings of our work is that a purely context-dependent hypermutation model can only partially explain the specificity of mutation hotspots. One could be worried that the additive nature of the model is to blame. We believe that position-dependent effects, i.e. depending on the position along the gene, could play an important role. In this revision we now learn a full 5-mer model. However, only a few 100s of the 1024 possible 5-mer are found in large enough quantities to quantify their mutabilities. When they do, the 5-mer context is often enough to specify the position of the mutation. The full 5-mer model performs better than the additive one, but this is due to the fact that the 5-mer acts as a proxy for position, consistent with our interpretation that position effects are important. We have now extended this discussion and have added a plot showing that the discrepancy is not due to the additive nature of the model.

Minor comments/typos

- Abstract: “and its modular structure can investigate models” - reword. It reads as if the modular structure is the researcher.

We have reworded to say “and its modular structure can be used to investigate models”.

- Page 3 Figure 1 caption: “Each segments gets trimmed a” - “segments” should be “segment”

Thank you. We have corrected this.

- Figure 2: The deletion and insertion distributions are plotted but it is not clear for which V/J gene. Is it combined for all genes? How does it vary between genes?

It is averaged over all genes weighted by the gene usage. For the insertion profile we assume a universal model for all genes (although IgoR can be configured to encode such dependencies, see new SI Table S1). For deletions we learn different models for each gene. We now state so clearly in the caption and we have added the deletion profiles gene by gene to the SI (Fig. S19).

- Page 4 Figure 2 caption: “s (b) V and (c) J genes.” - switch V and (b) and J and (c).

We agree that the switched order would be better. Unfortunately, the publisher requests this order in calling sub-figures (figure label must precede description).

- Page 6: “We found that the inference was highly accurate for datasets of 105 sequences and an error rate set to its typical experimental value, 10^{-3} (Fig. 3A and b), and was not affected by overfitting.” - can the authors supply evidence that this was not affected by overfitting?

The DKL presented in Fig.3c for artificial data and in the SI for real data is a measure of cross-validation error. Since it is low (below 1 bit) compared to the total entropy (50 bits) we have a small amount of overfitting. We have now clarified this.

Reviewer #3 (Remarks to the Author):

In this manuscript, Marcou et al. present “IGoR”, a novel tool for estimating and characterizing features of expressed antibody/B cell receptor and T cell receptor repertoires from high-throughput sequencing data. The authors present statistical assessments of their tool using simulated and real data (in both gDNA and RNA contexts), also offering direct comparisons to two additional existing tools. Although I believe the manuscript and analyses are technically sound and reach the bar for publication in Nature Comm., I would not be willing to recommend it for publication in Nature Communications in its present form. However, I believe just minor edits are required that provide more thoughtful background for motivating the need for such tools, as well as more thorough descriptions of their analyses and results in the context of the points raised in their discussion, so that the reader can better understand what is being presented. Please find my comments/questions below.

We thank the reviewer for his/her positive evaluation of our work and his/her detailed comments. Below we address them one by one. We have marked all the changes in the manuscript in red.

General Comments

1) In general, I found the level of detail, the layout of particular sections, and overall presentation of the results to be a little disjointed at times, jumping from one analysis to the next. For example, while I understand that IGoR is designed for use in both BCR and TCR datasets, at times it is not always clear to the reader when a distinction is made between deciding to present data on TCR vs. BCR datasets, or if both types of datasets were used to test all features of the software (excluding SHM).

We apologize for the confusion. We have now tried to make it clear when we are presenting results for TCRs and when for BCRs. We have added labels to the figures and we now try to be clear about the dataset in each question. For the first part of the manuscript (before the SHM section), we work with either a TCR model or a “naive” BCR model (without hypermutations).

And in some cases when both TCR and BCR data are tested and presented on, this is done in different ways (i.e., using different plots between the two types of repertoire datasets, or only showing results from 1 of them?; e.g., Figures S2 and S3, or comparison of 130bp IGH vs. 60 bp TRB?).

With the exception of Fig. 3 (and Fig. 5 which is for SHM) we have tried to show results for either the TCR or BCR and show the other receptor in the SI. Our goal was to avoid redundancy, at least in the main manuscript:

- In Fig. 3 and Fig. S2 we only showed results for TCR because the reads are shorter, which makes the problem more challenging.
- In Fig. 4 A we show both TCR and BCR;
- in Fig. 4B we show the BCR example and the TCR is plotted in SI Fig S6;
- in Fig 4D we use the TCR because the DJ exclusion rule does not exist in BCRs;
- and in Fig. 4C we show BCRs because we wanted to compare our results to Partis, which was developed for BCRs.

- Fig.S3 now includes both TRB and IGH.

Further to this point, many of the supplementary figure legends and results sections pertaining to them are sparse in detail, to say the least, at times lacking even axis labels (e.g., Fig. S17).

Thank you for pointing this out. We have now added axes to Fig. S17 and have reworked all SI figure captions.

This makes take home messages difficult to understand at times. More thoughtful attempt to link the motivation for this work to the results, and in turn to more detailed discussion of its potential application would go a long way to improving the clarity and impact of this manuscript.

We have tried to improve the overall clarity following the reviewer's suggestions. We also have extended the discussion of the potential applications of the method in the discussion.

Minor Comments

1) The term "scenario" is introduced in the introduction section, but this should be explained in more detail to the reader, especially given it is one of the metrics used to assess performance, and it mentioned many times throughout the manuscript.

We have clarified what we mean by "scenario" in detail in the introduction.

2) The "recombination of genomic segments" is not necessarily a "stochastic" process. This should be clarified.

We have added a sentence to clarify what we mean by the fact that the receptor generation process is stochastic. Each receptor can be produced by recombining pairs or triplets different V, (D) and J genes and with a different number of deletions or insertions at each junction. A specific gene choice does not predetermine a given number of deletions or insertions. Each realization of the receptor generation process that includes the same two or three genes can produce a different receptor, and conversely, each receptor can be generated using different genes and numbers of insertions and deletions. For these reasons the generation process is not deterministic but it is stochastic.

3) Fig. 1a uses IGH as an example, whereas Fig. 1b uses TCR. Not sure whether this is intentional. In some ways it speaks to the general comment on clarity above.

Thank you for pointing this out. We have changed Fig. 1b to use BCRs also.

4) Several abbreviations used throughout are not defined; e.g., "V", "D", "J", and "IGH".

We have now defined these abbreviations.

5) "By contrast, V and J gene usage varied moderately but significantly across individuals, and even more across sequencing technologies, suggesting possible primer-dependent biases (Fig. S4, see also Fig. S17 for IGH D-J gene usage)." This is an interesting observation, though not surprising given previous studies. Did you test whether this variation influences IGoR performance. In other words, are certain

libraries/techniques/individual repertoires better suited for IGoR? If so, why might this be?

No, the choice of technique, sequencing platform, repertoire does not influence IGoR's performance, in so far as the data is properly acquired and pre-processed and does not have biases in the acquisition process. We have now added a note about the importance of data quality control. IGoR uses data from different platforms as input. Knowing the details of how the data was acquired and whether it has any limitations will help interpret the data IGoR produces. For example, Illumina sequencing is less prone to indel errors than other platforms, and barcoding the data helps cluster unique sequences. We now note explicitly in the discussion that IGoR can be used with any platform.

6) "The maximum-likelihood scenario is not the correct one in 72% of IGH sequences and 85% of 60bp TRB sequences."
Should this read "72% of 130 bp IGH sequences..."?

Yes, thank you. We have corrected this sentence.

7) "For an error rate of 10⁻³, ~ 5000 unique out-of-frame sequences (which can be obtained from less than 2ml of blood with current mRNA sequencing technologies [14]) were sufficient to learn an accurate model of TRB (Fig. 3c)..."

Did you not test this explicitly for BCR? The sentences following the one quoted above suggest you have (as does Figure 3, S5), but again the results are presented in a confusing way.

Thank you for pointing this out. We have now removed the mention of hypermutations. We tested this point explicitly on TCRs and naive BCRs without a hypermutation model. Figure S5 is for TCRs. The difference between TCRs and naive BCRs is only in the type of gene, insertion and deletion distributions used. Since mixing the two is confusing, we chose to present this result only for TCRs.

8) Why were the read lengths used for TCR and BCR datasets selected? Does read length influence the performance of IGoR? Did you test? If not, some rationale for choosing the read lengths used seems warranted.

The synthetic read lengths were chosen to exactly mimick the experimental read lengths. These are typical read lengths we have been working with from hiSeq platforms. We have now added a sentence to explain our choice. IGoR's runtime decreases with longer read lengths. Identifying the correct V gene becomes easier, since longer read lengths provide more V sequence and one can more easily discard possibly V gene candidates. As a result the performance of other software is expected to become better for V usage and to converge with that of IgoR. However, since the insertion and deletion profiles cannot be identified in a unique way, IGoR's ability to learn these profiles (and other software's inability to do so) is not expected to change with read length. We also chose these read lengths are typical but hard examples, since longer read lengths would make the problem easier.

We now discuss these points in the main text.

9) In Figure S8, from your comparison of IGoR to MiXCR and Partis you state: "Gene usage is mostly consistent between methods."

This doesn't seem to be necessarily true? For many IGHV genes, the data presented

seems to indicate a good degree of variability between software packages (e.g., IGHV3-21 and IGHV3-30). Do the authors believe this is not relevant? If not, why?

We have now rerun the comparison using the same germline gene databases. Since Partis and MiXCR ship with their own databases we initially used the provided ones. The new comparison (Fig. S8) shows the gene usage is more similar between methods, although some discrepancies remain. We have removed the problematic sentence.

Were IGoR and MiXCR at least compared for TCR analysis, even though Partis can't be used for TCR?

We have performed the comparison on 60bp reads, but MiXCR discarded a very large number of sequences because of the short read length (60bp), which probably put most sequences below its tolerance threshold. We prefer not to include this comparison in the SI since the comparison is not favorable for MiXCR, nor is it really fair, as increasing read length a bit, or adjusting thresholds could change the result dramatically.

10) You mention the genomic data is a requisite for IGoR, and even comment that IGoR can be applied to any species for which genomic data is available. But what about species for which genomic data are available but germline databases are largely incomplete? Would you expect IGoR to perform less well in certain species, strains, or populations?

This is a very interesting question, which raises possible avenues for IGoR's future development. Currently IGoR requires genomic data from the species or from a closely related species. However, the type of probabilistic approach IGoR uses can be used to identify new alleles, so one could imagine starting from a largely incomplete genome or from a genome from a related species, and learning the genomic segments. Currently, IGoR does not have this option but it is an interesting future development. We now mention this direction in the discussion.

11) "Once a recombination model is learned for a given locus, IGoR can generate arbitrary numbers of synthetic sequences with the same statistics...". Is this referring to models for a locus within an individual, or more generally, in a species or population? One certainly seems more feasible than another based on what we understand from existing data. And it seems too early to make this point in such a definitive way, especially primarily based on the data presented here.

We apologize for the confusion. Technically, we meant locus within a given individual. However, in practice we have found that the probability of generating a given receptor is quite stable between individuals. Gene usage shows the greatest variability between people (and it is still quite small), however it has a relatively minor contribution to the overall probability of generating a sequence. To be very careful about gene usage differences between individuals that can depend on the HLA type etc, we (Pogorelyy et al 2017) and others (e.g. Emerson et al 2016) have found it useful to perform the analysis per V-J class. Within this break up, different individuals are well described by the same statistics. It is also possible to re-learn the gene usage for each dataset with IGoR, while keeping the insertion and deletion profiles to a universal value. We have now expanded this paragraph to make this clear.

12) How do you reconcile these two statements in the discussion?:
"and thus dispense with the need of a healthy control cohort",
"Its detailed analysis of recombination scenarios further reveals that, even with a

perfect estimator, the scenario is incorrectly called in more than 70% of sequences”???

These seem to conflict with one another.

Again, we apologize for the confusion. These two points are unrelated. We have rewritten the discussion to state them more clearly.

- The first citation refers to the point that, for each receptor sequence, IGoR gives the probability with which it was generated. This estimate is expected to be accurate. Since some sequences are much more likely to be generated than others, this estimate tells us whether we should be surprised to see a sequence in a given repertoire. If the sequence has a high probability to be generated and is found in many individuals, we should be less surprised than for a sequence that is very unlikely to be generated and is still found in many individuals. The latter one may be shared between individuals for functional reasons.

- The second citation makes the point that we CANNOT say exactly how a given sequence was generated. Neither we, nor anyone else, can say with certainty “TdT inserted 5 nt”. This is a statement about our uncertainty about the particular scenario that leads to a particular sequence, which follows both from the stochastic nature of recombination, and from the fact that several scenarios lead to the same sequence (convergent recombination).

The two statements are not contradictory. It is conceptually possible to estimate the probability that an event (here, a particular rearranged sequence) occurred, without knowing the exact sequence of causes (here, the recombination scenario) that led to it. This is exactly what IGoR does, by summing over all possible scenarios, and not presuming to know which one is correct.

REVIEWERS' COMMENTS:

Reviewer #1 (Remarks to the Author):

Let me begin by saying that IGoR is a very important work and definitely deserves publication and attention. The fact that the authors have been able to enable inference under an arbitrary graph structure encoding dependencies is a remarkable technical achievement and will form the building blocks of many future studies. There is also a lot of hard and clever work to take this concept and make it applicable to large data sets.

I am also happy to report that this software is now in what I would consider a usable state for others. The authors have made many improvements in this regard. We've continued to have a productive dialog, and I've offered a few more pull requests, which has resulted in improved documentation.

I am not going to ask for any further changes to the manuscript. However, I think that given how awesome this new software is, the paper still has some surprising aspects.

1. Regarding IGoR and sequences under selection the message is a still little mixed. On one hand, IGoR is designed to calculate generation probabilities of pre-selection sequences, but we hear

> nothing keeps IGoR from processing productive sequences in the analysis mode, either to call scenarios, hypermutations, or to estimate sequence generation probabilities

The paper now includes a validation in which a selection step is implemented to achieve a given CDR3 length distribution. From this analysis we hear

> adding selection to the dataset does not affect performance

This is a pretty strong statement from a fairly "light" validation (many things are expected to change under selection, not just CDR3 length), especially given that the authors are world-experts on exactly this process:

<http://dx.doi.org/10.1073/pnas.1409572111>

<http://dx.doi.org/10.1098/rstb.2014.0243>

2. Hypermutations form the main challenge for BCR annotation, and IGoR models hypermutation, yet the validation doesn't include hypermutation.

That is surprising, indeed surprising enough that it led me to misunderstand the previous version of the paper, which I assumed used hypermutation for the validation.

The challenge from hypermutation will never go away as technology improves. However, in this paper an important source of the uncertainty comes from short read lengths, which will go away as technology improves.

3. As I mentioned above, one of the truly remarkable things about this software is that it can do inference on models with arbitrary dependencies. It would be neat to see this functionality shown off in this paper by directly comparing different model formulations.

Details

169: I am guessing the authors mean rejection sampling rather than importance sampling -- importance sampling results in a weighted sample. I could be wrong though.

Sometimes we see "TCR beta" and sometimes "TCR β ", and sometimes TRB. The first two are clearly equivalent, but is there a subtle distinction with the last?

Reviewer #2 (Remarks to the Author):

The authors have addressed all of my concerns and I think that the manuscript is ready for publication.

Reviewer #3 (Remarks to the Author):

I am generally satisfied with the revised version of this manuscript.

However, I have one additional, but relatively minor editorial suggestion. The authors may wish to think about incorporating, just to make sure their statements are clear to readers of their article.

They refer to "genomic data" several times. I think it is perhaps better to use the term "germline database". E.g., lines 285-286 "for which germline databases are available". The availability of "genomic data" is a relatively vague way to state this. There is genomic data available for lots of species, but this does not necessarily mean these species have usable and/or reliable germline databases. It's a nuanced point, but not irrelevant, considering the likely audience of this article. Further on this topic, regarding lines 372 and 373, there is really no such thing as "full genomic data" (or rather, no such thing as a complete germline database) for any species, at present. The sentence on lines 372-373 should probably be rephrased to reflect this.

REVIEWERS' COMMENTS:

Reviewer #1 (Remarks to the Author):

Let me begin by saying that IGoR is a very important work and definitely deserves publication and attention. The fact that the authors have been able to enable inference under an arbitrary graph structure encoding dependencies is a remarkable technical achievement and will form the building blocks of many future studies. There is also a lot of hard and clever work to take this concept and make it applicable to large data sets.

I am also happy to report that this software is now in what I would consider a usable state for others. The authors have made many improvements in this regard. We've continued to have a productive dialog, and I've offered a few more pull requests, which has resulted in improved documentation.

We thank again the reviewer for his positive assessment, and for helping us make the paper and software much better.

I am not going to ask for any further changes to the manuscript. However, I think that given how awesome this new software is, the paper still has some surprising aspects.

1. Regarding IGoR and sequences under selection the message is a still little mixed. On one hand, IGoR is designed to calculate generation probabilities of pre-selection sequences, but we hear

> nothing keeps IGoR from processing productive sequences in the analysis mode, either to call scenarios, hypermutations, or to estimate sequence generation probabilities

The paper now includes a validation in which a selection step is implemented to achieve a given CDR3 length distribution. From this analysis we hear

> adding selection to the dataset does not affect performance

This is a pretty strong statement from a fairly "light" validation (many things are expected to change under selection, not just CDR3 length), especially given that the authors are world-experts on exactly this process:

<http://dx.doi.org/10.1073/pnas.1409572111>

<http://dx.doi.org/10.1098/rstb.2014.0243>

We agree that this statement might be a bit too strong. One should be careful to distinguish the ability to call the correct scenario, which is intrinsically limited and is not very strongly dependent on the model structure, and the ability to infer the correct recombination statistics. The papers the reviewer refers to concern the latter task, for which the nature of the sequences (productive or unproductive) matters more.

We have added a sentence explaining this distinction in an effort to tone down the message as heard by the reviewer.

2. Hypermutations form the main challenge for BCR annotation, and IGoR models

hypermutation, yet the validation doesn't include hypermutation.

That is surprising, indeed surprising enough that it led me to misunderstand the previous version of the paper, which I assumed used hypermutation for the validation.

The challenge from hypermutation will never go away as technology improves. However, in this paper an important source of the uncertainty comes from short read lengths, which will go away as technology improves.

We agree that hypermutations make the annotation problem more difficult, and will not go away with technology. IGoR may still work in that case, but we wanted to validate it in a well-controlled setting.

3. As I mentioned above, one of the truly remarkable things about this software is that it can do inference on models with arbitrary dependencies. It would be neat to see this functionality shown off in this paper by directly comparing different model formulations.

Some of this functionality is already shown off in the current version. We have learned the TRB model by either assuming a full dependence between the 3 germline genes, or only between D and J. In addition, we have tested in silico a model structure in which the insertion profile depended on the choice of the V segment (Table S1). Although there the comparison was with repgenHMM, the results are identical with IGoR.

Details

169: I am guessing the authors mean rejection sampling rather than importance sampling -- importance sampling results in a weighted sample. I could be wrong though.

The referee is right. It is rejection sampling.

We have modified the text.

Sometimes we see "TCR beta" and sometimes "TCR β ", and sometimes TRB. The first two are clearly equivalent, but is there a subtle distinction with the last?

All are equivalent, as clearly indicated at the first mention of TRA and TRB.

We have tried to uniformize the notation by removing TCR beta and replacing by β . Same thing for delta and gamma.

Reviewer #2 (Remarks to the Author):

The authors have addressed all of my concerns and I think that the manuscript is ready for publication.

Reviewer #3 (Remarks to the Author):

I am generally satisfied with the revised version of this manuscript.

However, I have one additional, but relatively minor editorial suggestion. The authors may wish to think about incorporating, just to make sure their statements are clear to readers of their article.

They refer to "genomic data" several times. I think it is perhaps better to use the term "germline database". E.g., lines 285-286 "for which germline databases are available". The availability of "genomic data" is a relatively vague way to state this. There is genomic data available for lots of species, but this does not necessarily mean these species have usable and/or reliable germline databases. It's a nuanced point, but not irrelevant, considering the likely audience of this article. Further on this topic, regarding lines 372 and 373, there is really no such thing as "full genomic data" (or rather, no such thing as a complete germline database) for any species, at present. The sentence on lines 372-373 should probably be rephrased to reflect this.

We thank the reviewer for this good suggestion. We have made the suggested replacement of 'genomic data' by 'germline database', and have rephrased the sentence on lines 372-373.